# Finely manipulating room temperature phosphorescence by dynamic lanthanide coordination toward multi-level information security

Longqiang Li[1,2], Jiayin Zhou[1,2], Junyi Han[1,2], Depeng Liu[1,2], Min Qi[1,2], Juanfang Xu[1,2], Guangqiang Yin [1,2] ✉ & Tao Chen [1,2,3] ✉

Room temperature phosphorescence materials have garnered significant attention due to their unique optical properties and promising applications. However, it remains a great challenge to finely manipulate phosphorescent properties to achieve desirable phosphorescent performance on demand. Here, we show a feasible strategy to finely manipulate organic phosphorescent performance by introducing dynamic lanthanide coordination. The organic phosphors of terpyridine phenylboronic acids possessing excellent coordination ability are covalently embedded into a polyvinyl alcohol matrix, leading to ultralong organic room temperature phosphorescence with a lifetime of up to 0.629 s. Notably, such phosphorescent performance, including intensity and lifetime, can be well controlled by varying the lanthanide dopant. Relying on the excellent modulable performance of these lanthanide-manipulated phosphorescence films, multi-level information encryption including attacker-misleading and spatial-time-resolved applications is successfully demonstrated with greatly improved security level. This work opens an avenue for finely manipulating phosphorescent properties to meet versatile uses in optical applications.

Room temperature phosphorescence (RTP) is a fascinating optical phenomenon in which long-lived emission can last from several seconds to hours under ambient conditions after removing the excitation light source[1-5]. Due to its unique luminescence properties such as long luminescence decay lifetime and large Stokes shift[6-11], it has become a promising candidate for numerous applications, such as information encryption[12-16], bioimaging[17-20], and organic optoelectronics[21-23]. Compared with the most widely used fluorescent tags for information encryption[24-28], RTP materials exhibit highly concealable and unclonable for innovative multi-level information coding owing to their

additional time dimension and tunable optical properties (phosphorescence lifetime, color, and intensity)[29-35]. In recent years, inorganic afterglow materials with multimodal excitation properties have demonstrated the advantages of encryption and anti-counterfeiting with multi-dimensional information[36-39]. However, inorganic afterglow materials are mainly prepared by ion doping, suffering from a high preparation temperature, environmental pollution, and limited metal resources. To address these issues, researchers have turned to organic RTP materials for developing advanced information encryption. For instance, Tang et al. developed a high-level encryption system enabled

[1]Key Laboratory of Advanced Marine Materials, Ningbo Institute of Materials Technology and Engineering, Chinese Academy of Sciences, Ningbo 315201, China. [2]School of Chemical Sciences, University of Chinese Academy of Sciences, Beijing 100049, China. [3]College of Material Chemistry and Chemical Engineering, Key Laboratory of Organosilicon Chemistry and Material Technology, Ministry of Education, Hangzhou Normal University, Hangzhou 311121 Zhejiang, China. ✉e-mail: yinguangqiang@nimte.ac.cn; tao.chen@nimte.ac.cn

by the activation of visible RTP via the complexation of crown ether with K⁺ for conformation locking [40]. Recently, Chi et al. presented a highly concealed information encryption on the basis of near-infrared phosphorescence by stepwise Förster resonance energy transfer (FRET)[41]. Despite the impressive advancements that have been made, the on-demand manipulation of RTP properties to achieve desirable phosphorescent performance without any tedious preparation process remains a formidable challenge, which greatly limits its related applications in high-level information encryption and anti-counterfeiting. In addition, combining multicolor fluorescence and finely manipulated RTP for multi-level information coding to improve encryption security is still not well established.

Lanthanide ($Ln^{3+}$) metal ions such as $Eu^{3+}$ and $Tb^{3+}$ as iconic luminescence centers have been widely used for practical applications in the field of emissive sensors, information encryption, and anti-counterfeiting [36,42–48]. Since the $f$–$f$ transition of $Ln^{3+}$ species is a forbidden process, it is generally necessary to select organic ligands with high molar coefficients for sensitization to realize highly efficient emission, which is also called antenna effect [49,50]. The ligand-to-metal photosensitized energy transfer (PSET) process originates from the triplet state ($T_1$) of organic ligand to the higher excited states of $Ln^{3+}$ [51–53], which is a competing pathway for the radiation transition of organic triplet-state exciton to the ground state. Therefore, we

consider whether it is possible to finely manipulate RTP based on such a competitive mechanism via the integration of $Ln^{3+}$ acceptors with long-lived organic phosphorescent donors. In addition, according to the RTP emission mechanism, the wavelength of phosphorescence emission is always longer than the fluorescence emission. If an optical system with illusive long-wavelength fluorescence emission and short-wavelength phosphorescence emission could be realized, it will greatly satisfy various unexpected application scenarios such as attacker-misleading information encryption and multi-layer information output systems.

In this work, we present finely modulable RTP materials by using terpyridine phenylboronic acid (TPYBOH) as the bifunctional fluorogen, which acts not only as an organic phosphor but also as a chelating ligand toward $Ln^{3+}$. The pristine organic RTP material (TPYBOH@PVA) that exhibited an intense blue phosphorescence with an ultralong lifetime of 0.629 s could be facilely prepared by efficient B−O click reaction. On account of the excellent binding ability of TPY ligand toward metal ions, $Ln^{3+}$ ($Eu^{3+}$ and $Tb^{3+}$, etc.) were introduced into the pristine organic RTP system for finely manipulating the phosphorescent performance via PSET (Fig. 1a). Consequently, an excellent optical system with precisely manipulable performance was facilely realized by controlling the amount of $Eu^{3+}$ and $Tb^{3+}$ dopants. Moreover, the $Ln^{3+}$-manipulated RTP system (TPYBOH@PVA−Ln) reveals illusive

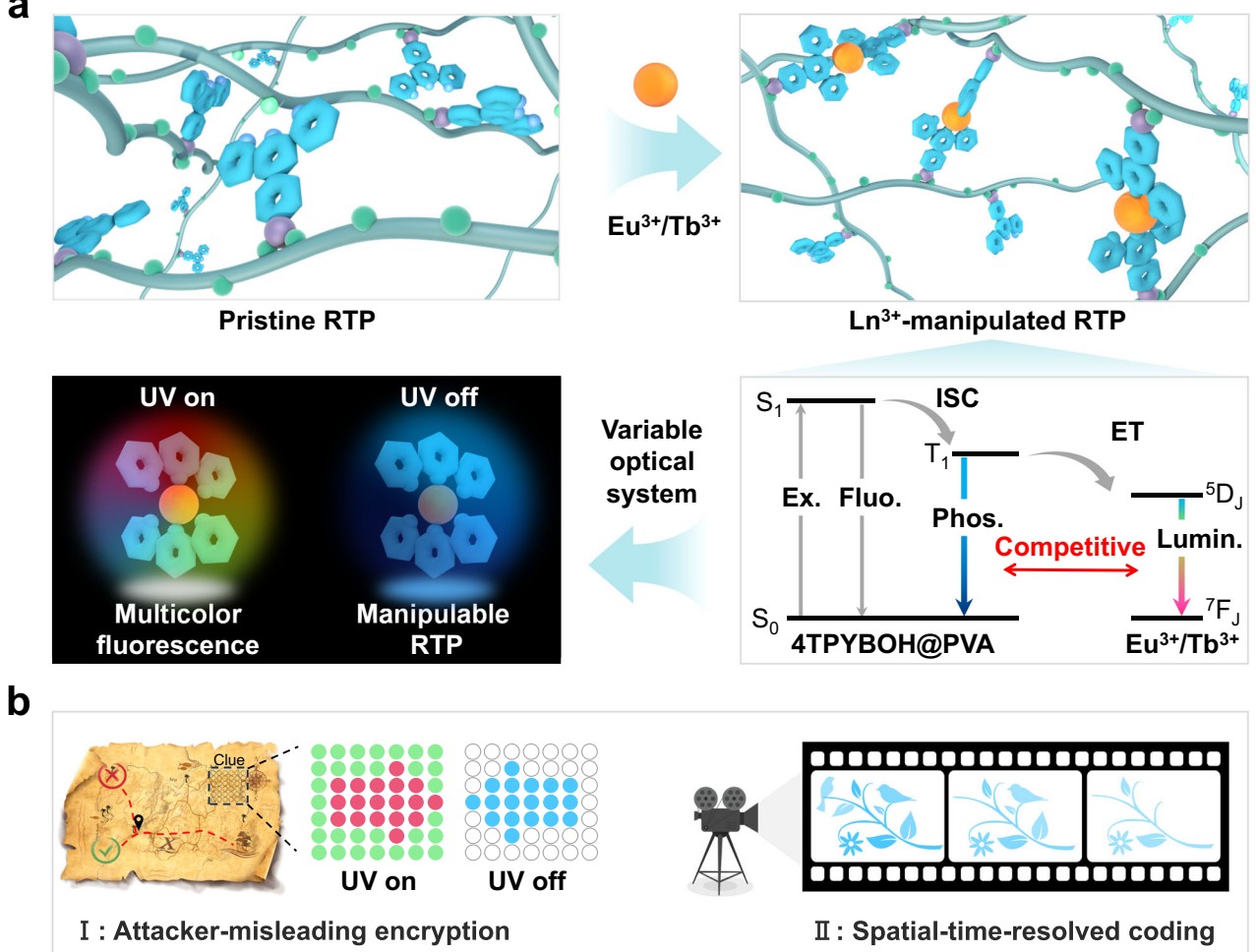

**Fig. 1 | The illustration of manipulating RTP properties and multi-level information encryption. a** The introduction of $Ln^{3+}$ ($Eu^{3+}$ and $Tb^{3+}$) into PVA-terpyridine phenylboronic acids (TPYBOH@PVA) for finely manipulating RTP properties via the ligand-to-metal photosensitized energy transfer process, the resulting $Ln^{3+}$-manipulated materials showing muticolor fluorescence and mutual blue phosphorescence. Ex.= excitation, Fluo. = fluorescence, Phos. = phosphorescence, Lumin. = luminescence. **b** The proof-of-concept demonstrations of the advanced multi-level encryption and anti-counterfeiting based on $Ln^{3+}$-manipulated RTP materials, revealing greatly enhanced security level.

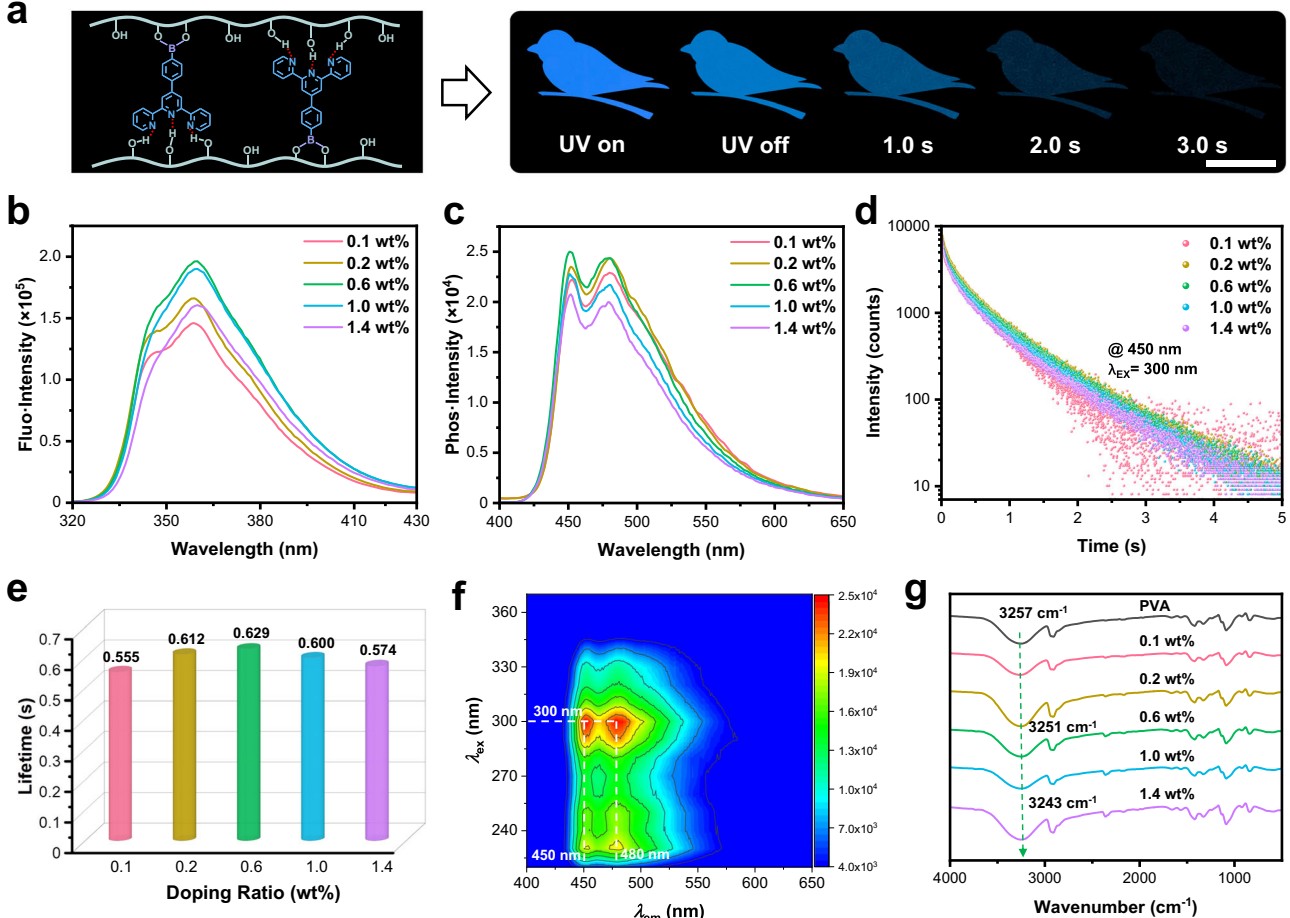

**Fig. 2 | Photophysical properties of the 4TPYBOH@PVA films. a** The illustration of chemical structure and hydrogen bonding of 4TPYBOH@PVA film and photographs of 0.6 wt% 4TPYBOH@PVA film taken under and after removing the 254 nm UV irradiation, scale bar = 1 cm. **b** Fluorescence spectra, **c** Phosphorescence spectra, **d** Phosphorescence-decay profiles and **e** Lifetime histogram (monitored at 450 nm) of 4TPYBOH@PVA films at different doping ratios of organic phosphor. **f** Excitation-phosphorescence mapping of 0.6 wt% 4TPYBOH@PVA film. **g** Attenuated total reflectance-Fourier transform infrared (ATR-FTIR) spectra of 4TPYBOH@PVA films at different doping ratios of organic phosphor.

long-wavelength fluorescence emission and short-wavelength phosphorescence emission, which is rarely reported but appealing for high-level information encryption[41,54]. Interestingly, the dynamic coordination between Ln[3+] and TPY units endows Ln[3+]-manipulated RTP materials with the excellent ability of reversible information writing and erasing. In addition, in sharp contrast to the previous triplet-to-singlet Förster resonance energy transfer (TS-FRET) method, the as-prepared Ln[3+]-manipulated RTP materials remain transparent under visible light, which is highly desirable for encrypting information with remarkable crypticity. As such, multi-level information encryption applications including attacker-misleading and spatial-time-resolved cryptographic systems are developed by taking advantage of these Ln[3+]-manipulated RTP materials (Fig. 1b).

## Results

### The fabrication and characterization of ultralong organic RTP films

For the sake of achieving finely Ln[3+]-manipulated RTP materials, we fabricated and optimized the pristine organic RTP material in the forefront step. Briefly, (4-([2,2′:6′,2″-terpyridin]-4′-yl) phenyl) boronic acid (4TPYBOH) was applied as organic phosphors to react with polyvinyl alcohol (PVA) by an efficient base-catalyzed B−O click reaction and followed by thermal annealing to afford ultralong organic RTP films. Strikingly, the 4TPYBOH@PVA film displayed an intense blue phosphorescence with a duration of ~3 s after removing 254 nm UV

irradiation (Fig. 2a). Thereafter, the optical properties of the 4TPYBOH@PVA film were finely optimized by varying the doping concentration of organic phosphor. The 4TPYBOH@PVA films exhibit blue fluorescence with an emission peak at 360 nm (Fig. 2b), exhibiting an optimal luminescence at 0.6 wt%.

Notably, the phosphorescence (delayed) spectra reveal dual-band emission at around 450 and 480 nm (Fig. 2c). The concentration-dependent RTP was consistent with prompt emission, exhibiting the strongest emission at a doping ratio of 0.6 wt%. It indicates that excessive organic phosphors cannot be effectively localized and confined by PVA matrix, resulting in energy dissipation and triplet quenching[9]. Based on dual phosphorescence emission, we conducted time-resolved emission-decay tests at 450 and 480 nm, respectively (Fig. 2d and Supplementary Fig. 1). After the fitting analysis, the lifetimes of 4TPYBOH@PVA (0.1–1.4 wt%) at 450 nm are 0.555, 0.612, 0.629, 0.600, and 0.574 s, respectively (Fig. 2e), which are comparable with the lifetimes measured at 480 nm (Supplementary Fig. 2). Furthermore, the transparency is slightly decreased as the doping amount of 4TPYBOH increased from 0.1% to 1.4%, but remains about 90% of transmittance in the visible region (Supplementary Fig. 3). In addition, no excitation-dependent RTP behavior is observed in the excitation range of 220–370 nm (Fig. 2f), indicating its isolated luminescent feature[55]. The above results prove that 0.6 wt% 4TPYBOH@PVA organic RTP film displays the best optical performance, and it is suitable for further performance manipulation by introducing Ln[3+].

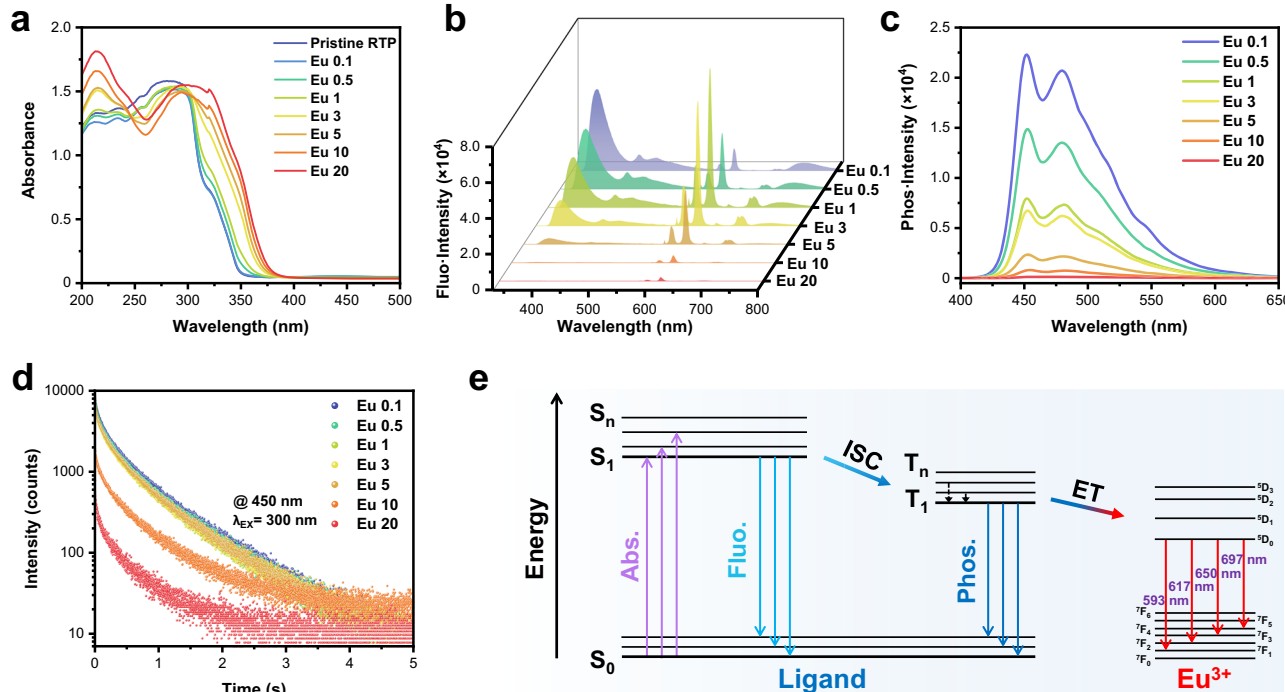

**Fig. 3 | Photophysical properties of the 4TPYBOH@PVA–Eu films. a** UV–vis absorption spectra, **b** Fluorescence spectra, **c** Phosphorescence spectra, and **d** phosphorescence-decay profiles of 4TPYBOH@PVA–Eu films that were prepared by being immersed in different $Eu^{3+}$(aq) (0.1–20 mg/mL). **e** Simplified Jablonski diagram for explaining RTP manipulation based on ET.

To gain more insights into the mechanism of RTP emission, control experiments and attenuated total reflectance-Fourier transform infrared (ATR-FTIR) spectroscopy were systematically employed. First, no RTP was observed for 4TPYBOH powders under ambient conditions (Supplementary Fig. 4), indicating the important role of covalently embedding boric phosphors into the polymeric networks for realizing ultralong organic RTP. Besides, we prepared 4TPYBOH@PVA film without the addition of alkali as a control experiment, namely, 4TPYBOH@PVA-C, which exhibited much weaker delayed emission and shorter lifetime compared with 4TPYBOH@PVA (Supplementary Fig. 5), further demonstrating the importance of B-O covalent bonding. To further investigate the importance of the boronic acid group, we utilized 4′-phenyl-2,2′:6′,2″-terpyridine (Ph-TPY) as the phosphor to integrate with the PVA matrix. As expected, the Ph-TPY@PVA exhibits a degraded optical performance in the fluorescence and phosphorescence emission (Supplementary Figs. 6–8). Moreover, the stretching absorption peak of the hydroxyl group is prominently shifted from 3257 to 3243 $cm^{-1}$ with the increase of the doping ratios (Fig. 2g), which proves the formation of strong hydrogen bonds between 4TPYBOH phosphor and PVA chains. Therefore, it can be concluded that covalent grafting by B-O click reaction and the geometrical confinement from multiple hydrogen bonding interactions immensely suppress the non-radiative dissipation of 4TPYBOH phosphor, leading to ultralong organic RTP emission.

**The controllable manipulation of RTP performance by $Ln^{3+}$ coordination**

With the optimized doping concentration and well-elucidated mechanism of 4TPYBOH@PVA film, we aim to finely manipulate its optical properties by dynamic coordination between organic phosphor and $Ln^{3+}$, which is highly desirable for multi-level information encryption. To be specific, europium ($Eu^{3+}$) and terbium ($Tb^{3+}$) were first introduced by immersing the pristine 4TPYBOH@PVA RTP film in a certain concentration of Eu $(NO_3)_3$ or Tb $(NO_3)_3$ aqueous solutions (0.1–20 mg/mL) for 5 min. As shown in Fig. 3a and Supplementary Fig. 9, the red shift and the rise of absorption spectra at 305–380 nm

due to the ligand-to-metal charge transfer proves that more $Ln^{3+}$ were introduced into the organic RTP system with the increase of the immersion concentration[56].

Moreover, X-ray photoelectron spectroscopy and ATR-FTIR spectroscopy were carried out to verify the coordination between TPY subunits and $Ln^{3+}$. Obviously, the binding energy of N 1$s$ was upshifted from 398.8 to 399.6 eV upon introducing $Ln^{3+}$ (Supplementary Fig. 10), attributed to the decrease of electron density on TPY after coordination[57]. In addition, an evident blue-shift of C=N stretching absorption further proved the coordination between TPYBOH and $Ln^{3+}$ (Supplementary Fig. 11). Interestingly, these films exhibit variable red or green fluorescence emission but remain blue phosphorescence emission (Fig. 3b, c and Supplementary Figs. 12, 18). It's worth noting that such illusive long-wavelength fluorescence emission with short-wavelength phosphorescence emission is rarely reported, which greatly satisfy more application scenarios.

As a control, the films (Ln@PVA) containing only PVA and $Ln^{3+}$ showed almost no fluorescence and phosphorescence emission (Supplementary Fig. 13). In addition, the powders of 4TPYBOH-Ln complexes only exhibited red and green fluorescence without long-lived RTP emission (Supplementary Fig. 14). Therefore, it can be concluded that the above mentioned long-wavelength fluorescence with short-wavelength phosphorescence emission occurs only in the three-component system. The fluorescence spectra of the three-component RTP films display characteristic peaks of $Eu^{3+}$ and $Tb^{3+}$ emission, demonstrating the successful introduction of $Ln^{3+}$ (Fig. 3b and Supplementary Fig. 12). The spectra of 4TPYBOH@PVA–Eu and 4TPYBOH@PVA–Tb RTP films follow the similar trend that the fluorescence intensity of $Ln^{3+}$ luminescence increased first and then decreased, along with maximum emission at a soaked concentration of 1 mg/mL. Take 4TPYBOH@PVA–Eu RTP materials for example, in addition to local emission of pristine material at 360 nm, it primarily has four characteristic peaks located at 593, 617, 650, and 697 nm, which can be assigned to the electron transitions of $^5D_0$-$^7F_J$ ($J$ = 1, 2, 3, and 4), respectively[58]. The local emission from the pristine

4TPYBOH@PVA at 360 nm is prominent at low doping concentrations of $Ln^{3+}$, while it becomes inconspicuous gradually as the $Ln^{3+}$ concentration increases over 1 mg/mL. As shown in the 1931 Commission Internationale de L'Eclairage (CIE) diagram, the fluorescence color gradually shifted to corresponding $Ln^{3+}$ emission upon the increase of doping concentration of $Ln^{3+}$ (Supplementary Fig. 15).

In phosphorescence spectra, the characteristic peaks of $Eu^{3+}$ and $Tb^{3+}$ still can be observed with a delay time of 0.05 ms (Supplementary Fig. 16), due to the long-lived $Ln^{3+}$ luminescence with lifetimes at hundreds of microseconds (Supplementary Fig. 17), which is hardly observable to the naked eye. Therefore, the delay time was adjusted to 100 ms to exclude the effect of $Ln^{3+}$ emission on the study of organic RTP. As shown in Fig. 3c and Supplementary Fig. 18, the RTP spectra exhibit characteristic emission peaks at 450 and 480 nm of the pristine 4TPYBOH@PVA film, indicating that the introduction of $Eu^{3+}$ and $Tb^{3+}$ has almost no influence on the energy level of lowest triplet state ($T_1$) of 4TPYBOH@PVA, which coincides with real observations showing blue phosphorescence. The RTP performance of $Eu^{3+}$- or $Tb^{3+}$-modulated films are all gradually degraded with the increased amounts of $Ln^{3+}$ (Fig. 3c, d and Supplementary Figs. 18–20). That is, the phosphorescent properties are highly dependent on the amount of introduced $Ln^{3+}$. Also, the lifetimes are shortened from 0.537 to 0.307 s for 4TPYBOH@PVA–Eu films and reduced from 0.509 to 0.185 s for 4TPYBOH@PVA–Tb films, respectively, upon increasing $Ln^{3+}$ concentration from 0.1 to 20 mg/mL. In addition, it is found that the fluorescence lifetimes monitored at 360 nm are shortened slightly from 3.07 to 2.08 ns upon increasing the $Eu^{3+}$ dopant (Supplementary Fig. 21).

Such manipulations of phosphorescence and fluorescence lifetimes may be attributed to two reasons: (i) the weakening of hydrogen bonding between organic phosphors and PVA chains after the coordination of TPY subunits with $Ln^{3+}$; (ii) increasing PSET from organic phosphors to $Ln^{3+}$ upon increasing $Ln^{3+}$ dopants. To figure out the manipulation mechanism, ATR-FTIR spectroscopy, powder X-ray diffraction and control experiments were systematically carried out. Except for the red shift of stretching vibrations of the pyridine ring (C=N, C=C), the stretching absorption of the hydroxyl group at around 3330 cm$^{-1}$ remains almost unchanged upon increasing $Ln^{3+}$ (Supplementary Fig. 11), indicating the introduction of $Ln^{3+}$ has a negligible effect on the overall hydrogen bonding. Besides, all the doped films

exhibit an obvious diffraction peak of PVA, along with a slight decrease in intensity as the amounts of $Ln^{3+}$ (Supplementary Fig. 22).

Furthermore, $La^{3+}$-doped RTP films were prepared and studied considering that the $La^{3+}$ had no available energy levels for energy transfer [59,60]. As expected, the RTP intensity and duration of delayed emission remain nearly constant (Supplementary Fig. 23), although an increase of $La^{3+}$ dopants may slightly weaken the hydrogen bonding (Supplementary Fig. 24). Undoubtedly, it proves that the PSET should be the dominant factor in performance regulation, rather than the weakening of hydrogen bonds. As a control experiment, polyacrylonitrile (PAN) was selected as a matrix to prepare modulable RTP films based on the confinement by electrostatic and dispersion interactions. Similarly, the RTP performance can also be finely manipulated by the introduction of $Ln^{3+}$ (Supplementary Figs. 25, 26), further validating the leading effect of the PSET on the manipulation.

Besides, cyclooctatetraene (COT), a well-known triplet quencher[61], was added to confirm whether the PSET from the singlet or triplet of the organic phosphors. As shown in Supplementary Fig. 27, the emission of $(4TPYBOH)_2$–$Eu^{3+}$ was greatly quenched after the addition of triplet quencher COT, implying that the predominant PSET process should be originated from the triplet of the organic phosphors. Furthermore, density functional theory calculations were performed to estimate the energy level of the organic phosphor and lanthanide complexes (Supplementary Fig. 28 and Supplementary Data 1). Compared with organic phosphor, the energy gap of lanthanide complexes was reduced greatly, indicating more easier for the lanthanide complexes to be excited. However, there is no significant difference in the singlet–triplet splitting energy ($\Delta E_{ST}$) for organic phosphors and lanthanide complexes. Through the above analysis, the PSET from the triplet state of 4TPYBOH phosphor to $Ln^{3+}$ should be a dominant pathway. In sum, such phenomena can be attributed to the competing relationship in the $Ln^{3+}$-doped RTP systems that more energy transfer occurs from the triplet state of 4TPYBOH@PVA to the excited state of $Ln^{3+}$, which leads to a decrease in the triplet emission of 4TPYBOH@PVA (Fig. 3e).

The above manipulation is more clearly reflected in the photographs (Fig. 4a). As the amount of $Ln^{3+}$ increases, the fluorescent color is gradually modulated from blue to bright red or green along with consistent blue phosphorescence. The 4TPYBOH@PVA–Eu and 4TPYBOH@PVA–Tb RTP films exhibit similar phosphorescent

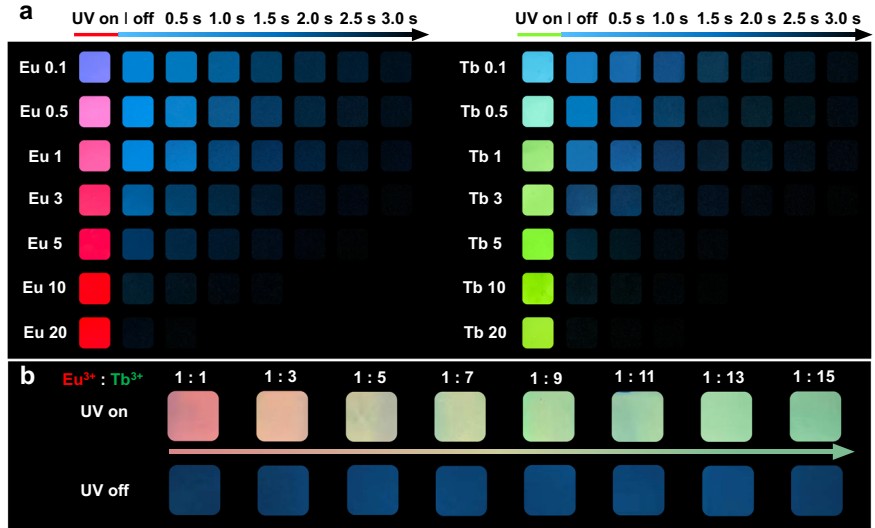

**Fig. 4 | The photographs of Ln$^{3+}$-manipulated RTP materials. a** The photographs of 4TPYBOH@PVA–Eu (left) and 4TPYBOH@PVA–Tb (right) RTP films that were prepared by being soaked in different Ln$^{3+}$ concentrations (0.1–20 mg/mL) taken under and after removing the 254 nm UV irradiation. **b** The photographs of bimetallic Eu$^{3+}$ and Tb$^{3+}$-manipulated RTP materials with the increase of the ratio of Tb$^{3+}$ taken under and after removing the 254 nm UV irradiation, the total Ln$^{3+}$ concentration is 1 mg/mL.

durations when immersing in the same concentration, suggesting such optical manipulation depends on the ratio of Ln³⁺ dopant. Notably, the fluorescence color remains nearly unchanged while the phosphorescence continues to diminish since the soaking concentration increased to 5 mg/mL. Moreover, we simultaneously introduced $Eu^{3+}$ and $Tb^{3+}$ to jointly manipulate the optical performance (Fig. 4b and Supplementary Fig. 29). As shown in Supplementary Fig. 29, the fluorescence spectra of the bimetallic-manipulated films show characteristic emission peaks of $Eu^{3+}$ and $Tb^{3+}$, demonstrating the successful introduction of $Eu^{3+}$ and $Tb^{3+}$ simultaneously. Along with increasing the proportion of $Tb^{3+}$, the fluorescent color gradually changes from red to yellow, and finally to opal green, while the phosphorescent performance remains unchanged owing to the constant concentration of total $Ln^{3+}$ (Fig. 4b).

Strikingly, white light emission with the CIE coordinate of (0.30, 0.32) was successfully achieved by finely adjusting the dopants of $Eu^{3+}$ and $Tb^{3+}$ with a more elaborate preparation process (Supplementary Figs. 30, 31). Moreover, these $Ln^{3+}$-manipulated RTP films remain transparent features (Supplementary Fig. 32), which is highly desirable for practical applications in information encryption and anti-counterfeiting with enhanced crypticity. The above results demonstrate that a desirable RTP system with finely modulable fluorescence, phosphorescence intensity and lifetime can be efficiently achieved by controlling the amount of $Ln^{3+}$.

To investigate whether the optical properties could be manipulated by the other lanthanide metals, we further introduced $Sm^{3+}$, $Dy^{3+}$, $Pr^{3+}$, $Ho^{3+}$, and $Tm^{3+}$ into the pristine organic RTP system. As shown in Supplementary Fig. 33, the RTP emission can also be finely modulated by doping these $Ln^{3+}$, in addition to relatively smaller changes in fluorescence emission compared with $Eu^{3+}$- and $Tb^{3+}$-doped films. On the one hand, phosphorescence spectra of these $Ln^{3+}$-manipulated RTP films all exhibit characteristic emission peaks at 450 and 480 nm, accompanied by significant decreases in the intensity and duration of RTP emission upon increasing the amounts of $Ln^{3+}$ dopants. It indicates that there is an efficient PSET between organic phosphors and $Ln^{3+}$. On the other hand, obvious characteristic fluorescence peaks of $Sm^{3+}$ and $Dy^{3+}$ emission can be observed, especially for $Sm^{3+}$-doped films that exhibit crimson fluorescence. However, no distinct fluorescence peaks of $Pr^{3+}$, $Ho^{3+}$, and $Tm^{3+}$ can be detected, which is attributed to the small energy gap between their excited and ground states, and the denser energy levels lead to severe non-radiative transition and energy dissipation[62,63]. Taken together, $Ln^{3+}$ ions with efficient PSET are expected to finely manipulate RTP properties, except for different abilities in the modulation of prompt emission.

## Demonstrations of multi-level information encryption and anti-counterfeiting

Considering the excellent modulable optical performance of these $Ln^{3+}$-manipulated RTP films, we decided to demonstrate their potential applications in high-level information security. Taking advantage of dynamic coordination between $Ln^{3+}$ and TPY ligand, we first demonstrated the cyclic information writing/erasing ability of these RTP films as shown in Fig. 5a. By spraying $Eu^{3+}/Tb^{3+}$ aqueous solutions through a designed mask on 4TPYBOH@PVA film, various encrypted patterns can be facilely printed. Expediently, these patterns were effectively erased by common chelator ethylenediaminetetraacetic acid disodium salt dihydrate (EDTA) due to the dynamic and reversible feature of $Ln^{3+}$–TPY coordination. Based on the writing/erasing procedure, pattern A (the pumpkin), pattern B (the panda), pattern C (the flower), and pattern D (the logo) are realized in turn. Importantly, owing to the reduction of phosphorescence intensity for $Ln^{3+}$-manipulated RTP films, these encrypted patterns exhibit an obvious change in contrast before and after removing the UV light, which manifests well-suitable for anti-counterfeiting applications. Furthermore, these patterns have an additional time dimension compared with the most widely used fluorescent tags, leading to a high-security level for verifying the

authentic commodities. Besides, we combine the Eu 5- and Eu 20-manipulated RTP materials to achieve an anti-counterfeiting pattern that hides information with similar fluorescence and displays target information by phosphorescence (Supplementary Fig. 34). Under UV light, the pattern was not readable, no useful information can be obtained due to similar fluorescence of Eu 5- and Eu 20-manipulated RTP materials. Whereas, a vivid spider-man pattern begins to appear after the removal of UV light.

In order to further elucidate their great potential in multi-level encryption, we designed and fabricated a dot matrix that provides misleading information by similar fluorescence and displays target information in the form of phosphorescence (Fig. 5b, Supplementary Fig. S35, and Supplementary Movie 1). A disguised pattern of the right arrow that composed of 4TPYBOH@PVA-Eu and 4TPYBOH@PVA-Tb was designed as shown in Supplementary Fig. 35. Under UV light, the pattern transmits the information (turn right) while it conveys the opposite information (turn left) after turning off the UV lamp, demonstrating attacker-misleading information encryption. Only by obtaining the correct key can guide someone to the target direction.

To further improve the security level, we designed a spatial-time-dual-resolved encrypted pattern without introducing multiple phosphors, which reveals a greatly simplified preparation procedure. Leveraging the advantages of the finely manipulated optical performance in a single material system, a flower-and-bird anti-counterfeiting pattern painted by a series of 4TPYBOH@PVA-Ln with lifetimes and intensity gradients (Fig. 5c, Supplementary Fig. 36, and Supplementary Movie 2). Upon the UV light excitation, it displays a vivid flower-and-bird pattern with intense fluorescence colors. However, the color of the pattern turns to blue immediately and fades in spatial and chronological order after removing the UV irradiation. As a result, distinguishable spatial-time-resolved phosphorescence patterns can be observed in turn, yielding high-level anti-counterfeiting.

Encouraged by the above-established multi-level information encryption, we further developed a virtually unbreakable multi-layer encryption system for important information storage (Fig. 6a). Different from the reported encryption strategies, we developed an innovative method for camouflaging information by multicolor fluorescence and decryption in the form of delayed emission, yielding greatly improved security. Moreover, the important data can be further encrypted in a spatial-time-resolved way based on the finely modulated RTP system. In this way, the information storage capacity is greatly enlarged and the important data can be hidden in a multi-level way with a specific code for each layer, greatly reducing the risk of being hacked.

As a proof-of-concept, we designed and fabricated an encrypted dot matrix ($8 \times 5$) by using a series of 4TPYBOH@PVA-Eu and 4TPYBOH@PVA-Tb RTP materials (Supplementary Fig. 37). Relying on the excellent modulable performance of these $Ln^{3+}$-manipulated RTP films, the encrypted dot matrix can output multiple signals (Fig. 6b, c and Supplementary Movie 3). Each signal serves as a lock for each layer, and the decoded information acts as a password for each lock. The only way to get the important encrypted information is to obtain all the passwords. Under daylight, all messages are hidden and no information can be obtained due to the high transparency of these RTP materials (Supplementary Fig. 38). However, all dots are emissive under UV irradiation. For simplicity, we define the output of a blue or red emissive dot denoting 1 while green emissive and non-emissive dot representing 0. Under UV light, this system gives information of UCAS (abbreviation of University of Chinese Academy of Sciences) according to the ASC II (American Standard Code for Information Interchange) code. After the removal of UV light, all dots emit blue phosphorescence, endowing an invalid code. Whereafter, a new binary code can be decoded to NIMTE (abbreviation of "Ningbo Institute of Materials Technology and Engineering") after creasing UV light for 1.0 s. As time prolonged to 2.0 s, the binary code can be further deciphered as

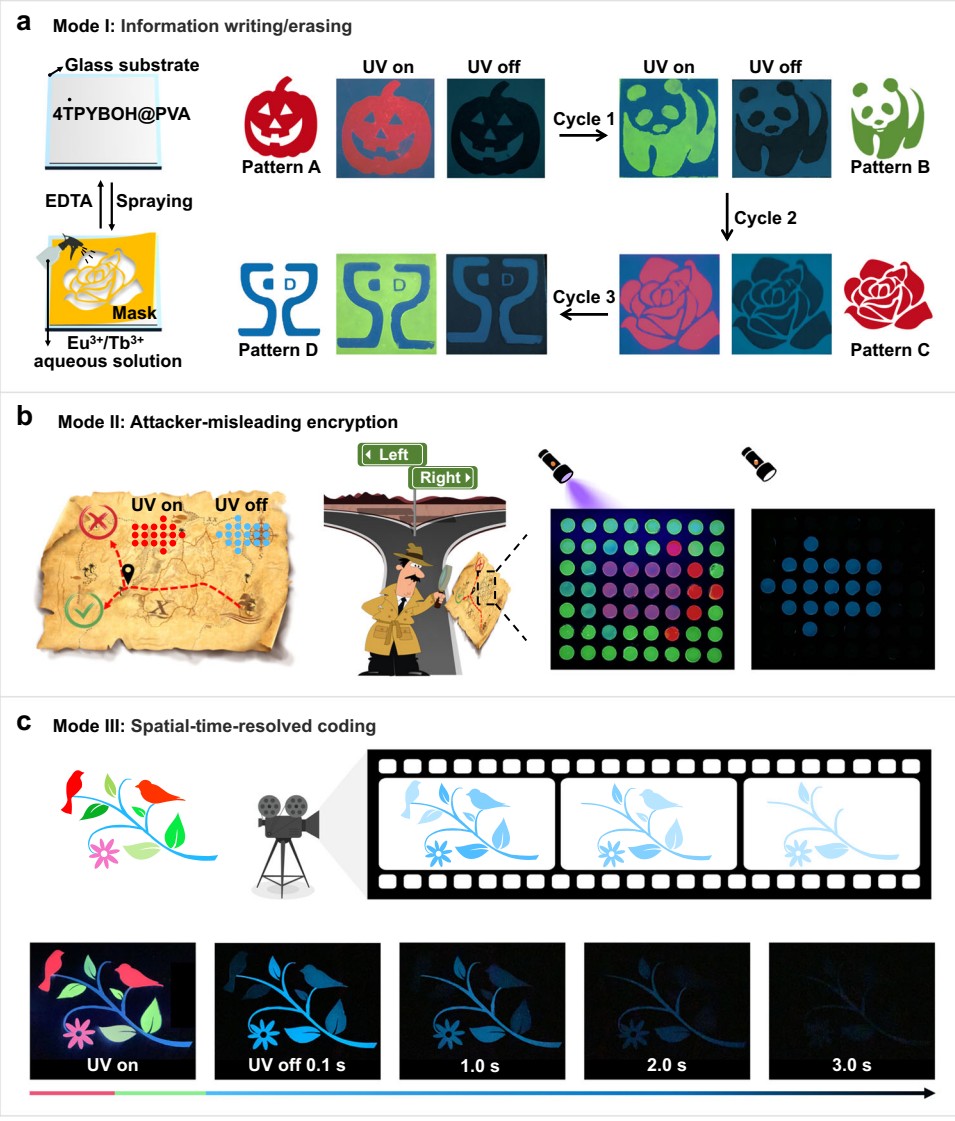

**Fig. 5 | Illustration of the multi-level information encryption and anti-counterfeiting. a** The schematic diagram of the information writing/erasing cycles and photographs of printed pattern taken under UV light on/off. **b** Demonstration of misleading information with disguised arrow pattern and displaying opposite direction upon creasing UV light. **c** Illustration of the spatial-time-resolved anti-counterfeiting by using different 4TPYBOH@PVA–Ln materials with lifetime gradients.

20044 (April 2004 witnessed the foundation of NIMTE) according to decimal numbers. In this way, whoever acquires the encrypted information not only requires obtaining a certain cryptographic principle but also needs getting the decoding rules. Such uncertainty of encoding rules and the diversity of output signals greatly avoids the possibility of decipherment and information leakage. As such, the multi-layer information encrypted storage system utilizing the above encrypted dot matrix also has a high-security level, which almost cannot be intercepted. Overall, these proof-of-concept demonstrations prove that these Ln³⁺-manipulated RTP films have great potential in multi-level information and greatly advance the development of related fields.

## Discussion

In summary, we have outlined an effective and feasible strategy to manipulate RTP performance by dynamic ligand–metal coordination, which enables multi-level information encryption. Ultralong organic RTP material was prepared by covalent embedding terpyridine phenylboronic acids into the PVA matrix, exhibiting a long-lived lifetime of 0.629 s with an intense blue phosphorescence lasting for 3 s. Thereafter, the optical properties including fluorescence emission, intensity, and duration of phosphorescence emission can be finely manipulated on demand by the introduction of dynamic $Ln^{3+}$ coordination to enable efficient ET. As a result, the phosphorescence intensity and lifetimes of these $Ln^{3+}$-manipulated RTP films were gradually decreased along with variable fluorescence emission from blue to green or red upon the increment of $Ln^{3+}$ dopants. Impressively, an illusive optical system with long-wavelength fluorescence emission and short-wavelength phosphorescence emission was facilely realized, leading to great promise in high-level information security. Relying on the excellent modulable performance of these $Ln^{3+}$-manipulated RTP films, multi-level information encryption systems including attacker-misleading, spatial-time-resolved, and the multi-layer encrypted dot matrix were successfully demonstrated. This work not only provides an ideal approach to finely manipulate organic RTP performance, but also broadens the applications of RTP materials in high-level information security and anti-counterfeiting.

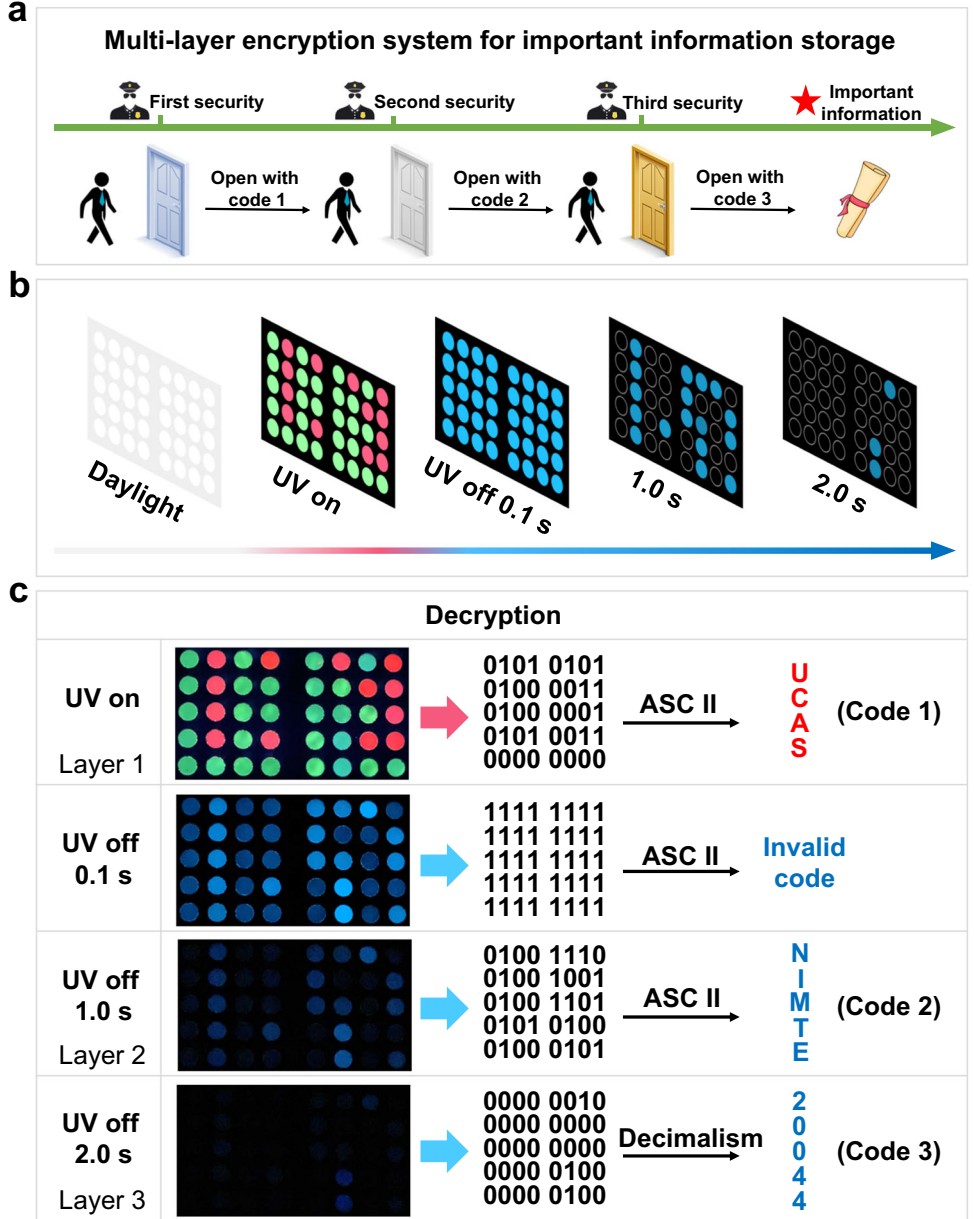

**Fig. 6 | Construction and decryption of a multi-layer encryption system for important information storage. a** The construction of a multi-layer encryption system for important information storage. **b** Multiple signals output by the encrypted dot matrix. **c** Illustration of the decryption process based on the definition principle and decoding rules.

## Methods

### Materials

PVA 1799 (PVA99), Sm(NO$_3$)$_3$·6H$_2$O (99.99%), Dy(NO$_3$)$_3$·5H$_2$O (99.99%), Pr(NO$_3$)$_3$·6H$_2$O (99.99%), Tm(NO$_3$)$_3$·6H$_2$O (99.99%), La(NO$_3$)$_3$·6H$_2$O (99.99%) were purchased from Energy Chemical. Ho(NO$_3$)$_3$·5H$_2$O (99.9%) and COT were purchased from Shanghai Macklin Biochemical Co. Ltd. PAN with an average Mw of 150 k was from Sigma-Aldrich. 4TPYBOH (97%) and (3-([2,2′:6′,2″-terpyridin]-4′-yl) phenyl)boronic acid (3TPYBOH, 98%) were purchased from Bidepharm. Benzaldehyde (>99.0%), 2-Acetylpyridine (98%), NaOH (97%), Eu(NO$_3$)$_3$·6H$_2$O (99.9%), Tb(NO$_3$)$_3$·5H$_2$O (99.9%), and Ethylenediaminetetraacetic acid disodium salt dihydrate (EDTA, 98%) were purchased from Aladdin Chemistry Co. Ltd.

### Preparation of RTP films

4TPYBOH@PVA films (0.1–1.4 wt%): the mixed solution of 500 mg PVA (99% alcoholysis degree) and 4TPYBOH powder (0.5, 1.0, 3.0, 5.0, 7.0 mg) in 7.0 mL H$_2$O and 1.0 mL aqueous NH$_3$·H$_2$O was reacted at 96 °C under stirring for 90 min to obtain the homogeneous precursor solution. Then, 2.0 mL of the homogeneous precursor solution was put on the glass slide (25 mm × 75 mm) at 65 °C for 2 h to remove H$_2$O and aqueous NH$_3$·H$_2$O. Finally, we can obtain transparent 4TPYBOH@PVA films (0.1–1.4 wt%) of about 100 µm in thickness.

Ph-TPY@PVA film: the mixed solution of 500 mg PVA (99% alcoholysis degree) and 3.0 mg Ph-TPY powder in 8.0 mL H$_2$O was heated at 96 °C under stirring for 90 min to obtain the homogeneous precursor solution. Then, 2.0 mL of the homogeneous precursor solution was put on the glass slide (25 mm × 75 mm) at 65 °C for 2 h to remove H$_2$O. Finally, we can obtain a transparent Ph-TPY@PVA film.

4TPYBOH@PVA–Eu/Tb films: first, Eu(NO$_3$)$_3$·6H$_2$O or Tb(NO$_3$)$_3$·5H$_2$O was added to H$_2$O to prepare the aqueous solutions with a concentration gradient (0.1, 0.5, 1.0, 3.0, 5.0, 10.0, 20.0 mg/mL). Then, the pristine RTP (0.6 wt% 4TPYBOH@PVA) films were immersed in different Eu$^{3+}$ or Tb$^{3+}$ aqueous solutions for 5 min and then heated at

65 °C for 2 h to remove $H_2O$. Finally, we can obtain the $Ln^{3+}$-doped RTP film with different doping amounts. $Sm^{3+}$-, $Dy^{3+}$-, $Pr^{3+}$-, $Ho^{3+}$-, $Tm^{3+}$-, and $La^{3+}$-doped RTP films were prepared by the same method with 4TPYBOH@PVA–Eu/Tb as described above.

## Data availability

The authors declare that the data supporting the findings of this study are provided within the article and its Supplementary Information file. All data are available from the authors upon request.

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

## Acknowledgements
This project was financially supported by the National Key Research and Development Program of China (2022YFB3204301 to T.C.), the National Natural Science Foundation of China (22205249 to G.Y.), Zhejiang Provincial Natural Science Foundation of China (LQ23B040002 to G.Y.), Ningbo Science and Technology Service Industry Demonstration (2020F042 to J.X.).

## Author contributions
G.Y. and T.C. conceived the project and revised the manuscript. L.L. was primarily responsible for the preparation and characterization, and wrote the manuscript. J.Z. guided drawing. J.H. performed theoretical calculations. D.L., M.Q. and J.X. gave valuable suggestions. All the authors discussed the results and proofread the manuscript.

## Competing interests
The authors declare no competing interests.

## Additional information

**Peer review information** : *Nature Communications* thanks Marcin Runowski and the other, anonymous, reviewers for their contribution to the peer review of this work. A peer review file is available.

