## [Peer Review File · Nature Communications]

Finely Manipulating Room Temperature Phosphorescence by Dynamic Lanthanide Coordination toward Multi-Level Information SecurityREVIEWER COMMENTS

Reviewer #1 (Remarks to the Author):

In this paper, the author reports a strategy to manipulate organic RTP performance by introducing dynamic coordination between the organic phosphor and lanthanide (Ln^{3+}) metal ions. The organic phosphor of terpyridine phenylboronic acids possessing excellent coordination ability was covalently embedded into a polyvinyl alcohol matrix, leading to ultralong organic RTP. Then, the RTP performance, including intensity and lifetime, was controlled by varying the Ln^{3+} dopant.

However, the PVA-doped RTP system has been widely investigated. The RTP performance is quite normal. The coordination between Ln^{3+} ions to quench the RTP is expected. The photophysical mechanism is nothing new. As a result, both the novelty and significance are far from Nat's high standards. Commun. I do not recommend the acceptance.

Before publishing elsewhere, the following issues can be considered.

1. Why are only Eu^{3+} and Tb^{3+} used as dopants, and can these methods be extended to other lanthanide elements?
2. In Figure 3b, it is observed that when the doping ratio is $\text{Eu}^{3+}:\text{Tb}^{3+}=1:7$, the sample's emission is already close to white luminescence. Can authors fine-tune the doping ratio to achieve purer white light emission?
3. When the molecule coordinates with metal ions, will it weaken hydrogen bonding, which may affect its phosphorescent emission and shorten the afterglow lifetime?
4. Some details in the article should be described concisely, and duplicate diagrams, such as those already presented in the main text, do not need to be reiterated in the supplementary materials.

Reviewer #2 (Remarks to the Author):

The manuscript presents a feasible strategy to manipulate RTP performance by dynamic ligand-metal coordination finely. The intensity and duration of afterglow emissions can be well regulated on demand by introducing dynamic coordination to enable efficient energy transfer. Benefiting such modulable performance of these manipulated RTP films, multi-level information encryption including attacker misleading, spatial-time-resolved, and multi-layered cryptographic applications were fully demonstrated. These results sound interesting and can inspire other scientists working in this field. The manuscript is recommended for publication after the following revisions.

1. The ligand-to-metal photosensitized energy transfer is enabled by the coordination between TPYBOH and lanthanide ions. More detailed experimental data are required to prove the complexity.
2. The authors should provide the lifetime variations of the fluorescent emission at 360 nm upon

increasing the coordination ratio to confirm whether the ligand-to-metal photosensitized energy transfers from the singlet or triplet of the Tpy ligand.

3. DFT calculations should be provided to estimate the energy level of the organic phosphors and complexes.

4. Eu^{3+} and Tb^{3+} metal ions were utilized to manipulate the RTP properties. Can the RTP properties also be manipulated by the other lanthanide ions? What happens if other lanthanide ions such as La^{3+} and Tm^{3+} are applied for complexation?

5. In the demonstration part, the authors present several creative applications in multi-level information encryption and anti-counterfeiting. More discussions are needed on the advantages of such fluorescence-phosphorescence dual-mode encryption over the other optical encryption strategies.

Reviewer #3 (Remarks to the Author):

The manuscript (Ref: NCOMMS-23-56012) of the paper entitled "Finely Manipulating Room Temperature Phosphorescence by Dynamic Lanthanide Coordination toward Multi-Level Information Security", in my opinion can be considered for publication in the journal: "Nature Communications" after its minor revision. The article shows very interesting and novel results concerning new the use of organic lanthanide-based phosphorescent materials for real-world applications including anti-counterfeiting and optical coding. The work is very well executed, the results are novel and reliable (the postulated effects are well-proved), and the presented findings deserve to be published in this journal. However, before publication of the manuscript, the following points should be addressed:

- the Authors use the phrase "afterglow" thorough the manuscript, however, this expression is reserved for inorganic phosphors exhibiting emission from the traps (stimulated by temperature), and it is frequently mistakenly used for organic phosphorescent materials, which exhibit just long lasting emission due to the spin-forbidden character of the radiation processes – triplet-singlet emission (see the fundamental work of professor Hölsä, J. Persistent Luminescence Beats the Afterglow: 400 Years of Persistent Luminescence. *Electrochem. Soc. Interface* 2009, 18 (4), 42–45. <https://doi.org/10.1149/2.F06094IF>.)

- the Authors write: "luminescent lifetimes" but the correct expression is "excited state lifetime" or "luminescence decay time"

- the Authors write about the application of room-temperature phosphoresce for information encryption, but it is suggested to write something/compare with encryption and anti-counterfeiting with afterglow emission of lanthanide and d-block metal ions (see e.g. 10.1002/adfm.202307791; 10.1002/adom.202300600)

- in the next paragraph, the Authors write about the use of lanthanide ions in information encryption and optical sensors, but they disregarded the recent breakthrough achievements in this field (e.g. 10.1002/adma.202302749; 10.1002/adfm.202214663; 10.1002/adfm.202307791

- the Author write: “excited singlet state (S_1) of Ln^{3+} ” – which is incorrect in relation to most if Ln^{3+} ions. This is because, as in the case of the radiative transitions within Ln^{3+} ions, occur usually between the terms of higher multiplicity (including tetraplets, pentaplets, hexaplets, and so on) with different total spin momentum (forbidden 4f-4f transitions), and such radiative processes should be called luminescence. so more correctly, would be to say: “higher excited states of Ln^{3+} ” or “excited states of Ln^{3+} with higher multiplicity”.

- The above mentioned mistake also includes the energy level diagram in Scheme 1.

- change: “phosphorescence emission is always larger than the fluorescence emission.” To “phosphorescence emission is always longer than the fluorescence emission.”

Page 4; line 97-98; the Authors write: “which is rarely reported but appealing for high-level information encryption.” – provide the missing reference

- “As shown in Figure 2a and S9, the red shift and the rise of absorption spectra at 305-380 nm due to the charge transfer proves that more Ln^{3+} were introduced into the organic RTP system with the increase of the immersion concentration.” – I think that the authors meant “energy transfer” instead of “charge transfer”

- change: “phosphorescent spectra (emission)” to “phosphorescence spectra (emission)”

- “long-lived Ln^{3+} luminescence with lifetime at tens of milliseconds” - lifetimes of some Ln^{3+} (including Eu^{3+} and Tb^{3+}) in inorganic matrices are typically within few ms, and within hundreds of μs for organic compounds. From the decay curves in Figure S15, it is clear that the mentioned lifetimes are about 1 ms (not tens of ms)

- change: “lifetimes are decreased” to “lifetimes are shortened”

Point-by-point response to the comments

To Reviewer 1

General comments: In this paper, the author reports a strategy to manipulate organic RTP performance by introducing dynamic coordination between the organic phosphor and lanthanide (Ln^{3+}) metal ions. The organic phosphor of terpyridine phenylboronic acids possessing excellent coordination ability was covalently embedded into a polyvinyl alcohol matrix, leading to ultralong organic RTP. Then, the RTP performance, including intensity and lifetime, was controlled by varying the Ln^{3+} dopant. However, the PVA-doped RTP system has been widely investigated. The RTP performance is quite normal. The coordination between Ln^{3+} ions to quench the RTP is expected. The photophysical mechanism is nothing new. As a result, both the novelty and significance are far from Nat. Commun.'s high standards. I do not recommend the acceptance. Before publishing elsewhere, the following issues can be considered.

Response to the General Comment:

Thank you for your valuable suggestion. All of your questions have been seriously considered. We are grateful to you for pointing out our inadequate elucidation of the significance and novelty of this work. This is also the space where our manuscript can be further improved. Based on your comments, we have made substantial revisions and emphatically highlighted our novelty compared with the reported papers, which have significantly improved our manuscript. The point-to-point responses according to your valuable suggestions have been addressed substantially in the revised version of our manuscript and marked with a **yellow** background. We hope our revised version can be accepted by *Nature Communications*. The primary novelty and significance of our Ln^{3+} -manipulated RTP can be summarized as follows:

- 1. In terms of material design:** Indeed, PVA-doped RTP materials have been widely investigated attributed to the high tensile strength, excellent crystallinity and oxygen resistance of PVA. Even recently, there is still a lot of impressive works about PVA-doped RTP materials being continuously published (Nat. Commun., 2023, 14, 4720; Adv. Mater. 2023, 35, 2306501; Adv. Mater.2023, 35, 2210489, etc.). However, in the reported PVA-doped RTP systems, the fluorogens dopants only acted as phosphorescent emitters. **In sharp contrast, we integrate a bifunctional fluorogen (terpyridine phenylboronic) with the PVA matrix, which acts not only as an organic phosphor but also as a chelating ligand towards lanthanide ions.** By this means, the optical properties of the PVA-doped RTP materials can be finely modulated, providing more opportunities for practical applications.

- 2. From the point of view of performance modulation:** The elaborate modulation of RTP emission remains challenging in reported RTP systems and is mainly based on the introduction of fluorescent dyes for FRET. **In this study, we propose a novel strategy to finely manipulate optical performance by dynamic coordination, leading to multicolour fluorescence and elaborate RTP performance in intensity, lifetime and duration of delayed emission.** The optical properties including fluorescence and RTP emission are highly dependent on the amount of Ln^{3+} dopant. Importantly, the Ln^{3+} -modulated RTP materials remain transparent under daylight, which is highly desirable for information encryption but extremely difficult to achieve with FRET methods. Moreover, the Ln^{3+} -modulated RTP materials exhibit illusive long-wavelength fluorescence with short-wavelength delayed emission based on different luminescent mechanisms from previous RTP works.
- 3. From an application perspective:** In the previous works, the information encryption applications simply rely on the fluorescence to phosphorescence switches. **Differently, we present attacker-misleading information encryption based on information camouflage by multicolour fluorescence and decryption in the form of delayed emission, yielding greatly improved crypticity and security. Besides, spatial-time-resolved anti-counterfeiting can be facilely realized by leveraging the precise gradients of RTP emission.**
- 4.** The study of lanthanide luminescence has a long history. However, to the best of our knowledge, the ultralong RTP of lanthanide-based coordination systems has rarely been reported. In the present work, we systematically study the RTP emission and underlying mechanism of terpyridine-lanthanide-involved hybrid material.

As such, the material design, performance modulation, luminescent mechanism and cryptographic strategy of our work are completely different from these reported papers. We politely seek your understanding that our work is innovative and valuable for inspiring the development of fascinating RTP materials and advanced high-level cryptographic applications. We have highlighted and added more discussions on our novelty in the revised manuscript.

For example:

In Paragraph 2 of Page 3:

“In this work, we present finely modulable RTP materials by using terpyridine phenylboronic acid (TPYBOH) as the bifunctional fluorogen, which acts not only as an organic phosphor but also as a chelating ligand towards Ln^{3+} . The pristine organic RTP material (TPYBOH@PVA) that exhibited an intense blue phosphorescence with

an ultralong lifetime of 0.629 s could be facilely prepared by efficient B-O click reaction.”

In Paragraph 1 of Page 4:

Interestingly, the dynamic coordination between Ln^{3+} and TPY units endows Ln^{3+} -manipulated RTP materials with the outstanding ability of reversible information writing and erasing. In addition, in sharp contrast to the previous triplet-to-singlet Förster-resonance energy-transfer (TS-FRET) method, the as-prepared Ln^{3+} -manipulated RTP materials remain transparent under visible light, which is highly desirable for encrypting information with excellent crypticity. As such, multi-level information encryption including attacker-misleading and spatial-time-resolved cryptographic systems was developed by taking advantage of these Ln^{3+} -manipulated RTP materials (Fig. 1b).

In Paragraph 2 of Page 13:

“Based on the above-established multi-level information encryption, we further developed a virtually unbreakable multi-layer encryption system for important information storage (Fig. 6a). Different from the reported encryption strategies, we developed an innovative method for camouflaging information by multicolour fluorescence and decryption in the form of delayed emission, yielding greatly improved security. Moreover, the important data can be further encrypted in a spatial-time-resolved way based on the finely modulated RTP system. In this way, the information storage capacity is greatly enlarged and the important data can be hidden in a multi-level way with a specific code for each layer, greatly reducing the risk of being hacked. As a proof-of-concept, we designed and fabricated an encrypted dot matrix (8×5) by using a series of 4TPYBOH@PVA-Eu and 4TPYBOH@PVA-Tb RTP materials (Supplementary Fig. 37).”

Comment 1

Why are only Eu^{3+} and Tb^{3+} used as dopants, and can these methods be extended to other lanthanide elements?

Response to comment 1

Thank you for your suggestive comments. We rational select Eu^{3+} and Tb^{3+} as dopants considering their outstanding luminescent properties including narrow spectral lines, high quantum yields, and excellent photostability. According to your suggestion, we have also studied the optical modulation by introducing other lanthanide metal ions including La^{3+} , Sm^{3+} , Dy^{3+} , Pr^{3+} , Ho^{3+} and Tm^{3+} . It's found that Ln^{3+} ions with efficient PSET are highly desirable for finely manipulating RTP properties, except for different

abilities in the modulation of prompt emission. The corresponding fluorescence spectra, phosphorescence spectra, photographs and necessary discussions have been fully added to the revised manuscript. Please find them in **Paragraph 3 of Page 8, Paragraph 2 of Page 10, Supplementary Fig. 23** and **Supplementary Fig. 33** of the revised version.

In the revised manuscript:

In Paragraph 3 of Page 8:

Moreover, La³⁺-doped RTP films were prepared and studied considering that the La³⁺ had no available energy levels for energy transfer.^{59, 60} As expected, the RTP intensity and duration of delayed emission remain nearly constant (Supplementary Fig. 23), although an increase of La³⁺ dopants may slightly weaken the hydrogen bonding (Supplementary Fig. 24).

In Paragraph 2 of Page 10:

To investigate whether the optical properties could be manipulated by the other lanthanide metals, we further introduced Sm³⁺, Dy³⁺, Pr³⁺, Ho³⁺, and Tm³⁺ into the pristine organic RTP system. As shown in Supplementary Fig. 33, the RTP emission can be also finely modulated by doping these Ln³⁺, in addition to relatively smaller changes in fluorescence emission compared with Eu³⁺- and Tb³⁺-doped films. On the one hand, phosphorescence spectra of these Ln³⁺-manipulated RTP films all exhibit characteristic emission peaks at 450 nm and 480 nm, accompanied by significant decreases in the intensity and duration of RTP emission upon increasing the amounts of Ln³⁺ dopants. It indicates that there is an efficient PSET between organic phosphors and Ln³⁺. On the other hand, obvious characteristic fluorescence peaks of Sm³⁺ and Dy³⁺ emission can be observed, especially for Sm³⁺-doped film that exhibit crimson fluorescence. However, no distinct fluorescence peaks of Pr³⁺, Ho³⁺, and Tm³⁺ can be detected, which is attributed to the small energy gap between their excited and ground states, and the denser energy levels lead to severe non-radiative transition and energy dissipation.^{62, 63} Taken together, Ln³⁺ ions with efficient PSET are expected to finely manipulate RTP properties, except for different abilities in the modulation of prompt emission.

In the revised supporting information:

Supplementary Figure 23. (a) The photographs of La³⁺-doped RTP films (prepared by soaking 4TPYBOH@PVA in 1.0 and 10.0 mg/mL of La³⁺ aqueous solutions), the photos were taken under 254 nm UV excitation and at different time intervals after the removal of UV irradiation. (b) Phosphorescence spectra of corresponding La³⁺-doped RTP films.

Supplementary Figure 33. (a) The fluorescence and (b) phosphorescence spectra of Sm³⁺-, Dy³⁺-, Pr³⁺-, Ho³⁺-, and Tm³⁺-doped RTP films that were prepared by being immersed in Ln³⁺ aqueous solution (1.0 and 10.0 mg/mL). (c) The photographs of Sm³⁺-, Dy³⁺-, Pr³⁺-, Ho³⁺-, and Tm³⁺-doped RTP films that were prepared by being soaked in Ln³⁺ aqueous solution (1.0 and 10.0 mg/mL, respectively) taken under and after removing the 254 nm UV irradiation.

In the Experimental Section:

Materials. Polyvinyl alcohol 1799 (PVA99), $\text{Sm}(\text{NO}_3)_3 \cdot 6\text{H}_2\text{O}$ (99.99%), $\text{Dy}(\text{NO}_3)_3 \cdot 5\text{H}_2\text{O}$ (99.99%), $\text{Pr}(\text{NO}_3)_3 \cdot 6\text{H}_2\text{O}$ (99.99%), $\text{Tm}(\text{NO}_3)_3 \cdot 6\text{H}_2\text{O}$ (99.99%), $\text{La}(\text{NO}_3)_3 \cdot 6\text{H}_2\text{O}$ (99.99%) were purchased from Energy Chemical. $\text{Ho}(\text{NO}_3)_3 \cdot 5\text{H}_2\text{O}$ (99.9%) and cyclooctatetraene (COT) were purchased from Shanghai Macklin Biochemical Co. Ltd.

4TPYBOH@PVA-Eu/Tb films: First, $\text{Eu}(\text{NO}_3)_3 \cdot 6\text{H}_2\text{O}$ or $\text{Tb}(\text{NO}_3)_3 \cdot 5\text{H}_2\text{O}$ was added to H_2O to prepare the aqueous solutions with a concentration gradient (0.1 mg/mL, 0.5 mg/mL, 1.0 mg/mL, 3.0 mg/mL, 5.0 mg/mL, 10.0 mg/mL, 20.0 mg/mL). Then, the pristine RTP (0.6 wt% 4TPYBOH@PVA) films were immersed in different Eu^{3+} or Tb^{3+} aqueous for 5 minutes and heated at 65 °C for 2 hours to remove H_2O . Finally, we can obtain the Ln^{3+} -doped RTP film with different doping amounts. Sm^{3+} -, Dy^{3+} -, Pr^{3+} -, Ho^{3+} -, Tm^{3+} - and La^{3+} -doped RTP films were prepared by the same method with 4TPYBOH@PVA-Eu/Tb as described above.

Comment 2

In Figure 3b, it is observed that when the doping ratio is $\text{Eu}^{3+} : \text{Tb}^{3+} = 1:7$, the sample's emission is already close to white luminescence. Can authors fine-tune the doping ratio to achieve purer white light emission?

Response to comment 2

Thanks a lot for your helpful and valuable comments. According to your advice, we have tried our best to achieve white light emission. Obtaining white light emission requires a more elaborate doping ratio of Eu^{3+} and Tb^{3+} , which is difficult to realize with the previous method of direct soaking. Thus, we tried the other doping method by redissolving 4TPYBOH@PVA film, followed by finely adjusting the dopants of Eu^{3+} and Tb^{3+} . Ultimately, we successfully obtained the white light emission with CIE coordinate of (0.30, 0.32) although the preparation process is more complicated. The necessary discussions, preparation strategy and experimental data are fully added in the revised manuscript and supporting information. Please find them in **Paragraph 1 of Page 10, Supplementary Fig. 30 and Supplementary Fig. 31** of the revised version.

In the revised manuscript:

Along with increasing the proportion of Tb^{3+} , the fluorescent color gradually changes from red to yellow to opal green, while the phosphorescent performance maintains unchanged owing to the constant concentration of total Ln^{3+} (Fig. 4b). Strikingly, white light emission with the CIE coordinate of (0.30, 0.32) was successfully achieved by finely adjusting the dopants of Eu^{3+} and Tb^{3+} with a more elaborate preparation process (Supplementary Figs. 30-31).

In the revised Supporting information:

Supplementary Figure 30. (a) Fluorescence spectrum and (b) Corresponding CIE coordinate of 4TPYBOH@PVA-Ln (the molar ratio of Eu^{3+} and Tb^{3+} is 1:3 with a total mass of 1.5 mg, 3.0 mg, 4.0 mg, 5.0 mg and 6.0 mg, respectively), $\lambda_{\text{ex}} = 254 \text{ nm}$.

Supplementary Figure 31. The photographs of 4TPYBOH@PVA-Ln with white light emission were taken under and after removing the 254 nm UV irradiation.

In the Experimental Section:

4TPYBOH@PVA-Ln film with white light emission: The 8.0 mL homogeneous precursor solution of 0.6 wt% 4TPYBOH@PVA was heated at 100 °C for 12 hours. After all solvents have completely evaporated, 8 mL H_2O , the mixture of $\text{Eu}(\text{NO}_3)_3 \cdot 6\text{H}_2\text{O}$ and $\text{Tb}(\text{NO}_3)_3 \cdot 5\text{H}_2\text{O}$ (the molar ratio of Eu^{3+} and Tb^{3+} is 1:3, the total mass is 1.5 mg, 3.0 mg, 4.0 mg, 5.0 mg, 6.0 mg) were added into the dry 4TPYBOH@PVA and heated at 96 °C for 3 hours until thoroughly dissolved. Then, 2.0 mL of the above Ln^{3+} -doped precursor solution was taken and drop-casted on a glass slide (25 mm×75 mm). Finally, the glass slide was then heated at 65 °C for 2 hours to obtain Ln^{3+} -doped RTP film with white light emission.

Comment 3

When the molecule coordinates with metal ions, will it weaken hydrogen bonding, which may affect its phosphorescent emission and shorten the afterglow lifetime?

Response to comment 3

Thank you very much for carefully pointing out this important issue. Indeed, the hydrogen bonding between the tripyridine unit and PVA chains is bound to be weakened to some extent. To study the impact of such a weakening of hydrogen bonding on overall confinement, RTP emission and lifetime, attenuated total reflectance-Fourier transform infrared (ATR-FTIR) spectroscopy, powder X-ray diffraction (PXRD) and control experiments were systematically carried out. The experimental results reveal that the weakened hydrogen bonding has a very limited effect on the manipulation of the RTP properties. Please find them in the **Paragraph 3 of Page 8, Supplementary Fig. 11 and Supplementary Figs. 22-26** of the revised version.

In the revised manuscript:

Such manipulations of phosphorescence and fluorescence lifetimes may be attributed to two reasons: i) The weakening of hydrogen bonding between organic phosphors and PVA chains after the coordination of TPY subunits with Ln^{3+} ; ii) Increasing PSET from organic phosphors to Ln^{3+} upon increasing Ln^{3+} dopants. To figure out the manipulation mechanism, ATR-FTIR spectroscopy, powder X-ray diffraction (PXRD) and control experiments were systematically carried out. Except for the red-shift of stretching vibrations of the pyridine ring (C=N, C=C), the stretching absorption of the hydroxyl group at around 3330 cm^{-1} remains almost unchanged upon increasing Ln^{3+} (Supplementary Fig. 11), indicating the introduction of Ln^{3+} has a negligible effect on the overall hydrogen bonding. Besides, all the doped films exhibit an obvious diffraction peak of PVA, along with a slight decrease in intensity as increasing the amounts of Ln^{3+} (Supplementary Fig. 22). Moreover, La^{3+} -doped RTP films were prepared and studied considering that the La^{3+} had no available energy levels for energy transfer.^{59,60} As expected, the RTP intensity and duration of delayed emission remain nearly constant (Supplementary Fig. 23), although an increase of La^{3+} dopants may slightly weaken the hydrogen bonding (Supplementary Fig. 24). Undoubtedly, it proves that the PSET should be the dominant factor in performance regulation, rather than the weakening of hydrogen bonds. As a control experiment, polyacrylonitrile (PAN) was selected as a matrix to prepare modulable RTP films based on the confinement by electrostatic and dispersion interactions. Similarly, the RTP performance can be also finely manipulated by the introduction of Ln^{3+} (Supplementary Figs. 25-26), further validating the leading effect of the PSET on the manipulation.

In the revised Supporting information:

Supplementary Figure 11. ATP-FTIR spectra of PVA, 4TPYBOH@PVA, 4TPYBOH@PVA-Eu 1, and 4TPYBOH@PVA-Eu 10 RTP films.

Supplementary Figure 22. Powder XRD patterns of PVA, 4TPYBOH@PVA, 4TPYBOH@PVA-Eu 1 and 4TPYBOH@PVA-Eu 10 RTP films.

Supplementary Figure 23. (a) The photographs of La³⁺-doped RTP films (prepared by soaking 4TPYBOH@PVA in 1.0 and 10.0 mg/mL of La³⁺ aqueous solutions), the photos were taken under 254 nm UV excitation and at different time intervals after the removal of UV irradiation. (b) Phosphorescence spectra of corresponding La³⁺-doped RTP films.

Supplementary Figure 24. Powder XRD patterns of 4TPYBOH@PVA-La 1 and 4TPYBOH@PVA-La 10.

Supplementary Figure 25. Phosphorescence spectra of 4TPYBOH@PAN-Eu with different doping weights, the numbers represent 4TPYBOH@PAN doped with 0 mg, 1.0 mg, 2.0 mg and 5.0 mg $\text{Eu}(\text{NO}_3)_3 \cdot 6\text{H}_2\text{O}$, respectively.

Supplementary Figure 26. The photographs of 4TPYBOH@PAN-Eu RTP films at different doping weights of $\text{Eu}(\text{NO}_3)_3 \cdot 6\text{H}_2\text{O}$ were taken under and after removing the 254 nm UV irradiation.

In the Experimental Section:

Materials. Polyvinyl alcohol 1799 (PVA99), $\text{Sm}(\text{NO}_3)_3 \cdot 6\text{H}_2\text{O}$ (99.99%), $\text{Dy}(\text{NO}_3)_3 \cdot 5\text{H}_2\text{O}$ (99.99%), $\text{Pr}(\text{NO}_3)_3 \cdot 6\text{H}_2\text{O}$ (99.99%), $\text{Tm}(\text{NO}_3)_3 \cdot 6\text{H}_2\text{O}$ (99.99%), $\text{La}(\text{NO}_3)_3 \cdot 6\text{H}_2\text{O}$ (99.99%) were purchased from Energy Chemical. $\text{Ho}(\text{NO}_3)_3 \cdot 5\text{H}_2\text{O}$ (99.9%) and cyclooctatetraene (COT) were purchased from Shanghai Macklin Biochemical Co. Ltd. Polyacrylonitrile (PAN) with an average Mw of 150 k was from Sigma-Aldrich.

4TPYBOH@PAN-Eu films: The mixed solution of 500 mg polyacrylonitrile (PAN), 3.0 mg 4TPYBOH powder and $\text{Eu}(\text{NO}_3)_3 \cdot 6\text{H}_2\text{O}$ (0 mg, 1.0 mg, 2.0 mg, 5.0 mg) in 8.0 mL *N, N*-Dimethylformamide (DMF) was heated at 45 °C under stirring for 90 minutes to obtain the homogeneous precursor solution. Then, 2.0 mL of the homogeneous precursor solution was taken and drop-casted on a glass slide (25 mm×75 mm). Finally, 4TPYBOH@PAN-Eu films were obtained by heating the resulting glass slide at 50 °C for 4 hours to completely remove DMF.

Measurements. The spectra of ^1H nuclear magnetic resonance (NMR) were obtained by a Bruker AVANCE 400 MHz spectrometer. X-ray photoelectron spectroscopy (XPS) was carried out with AXIS SUPRA. Powder X-ray diffraction (PXRD) was performed on a Bruker D8 ADVANCE DAVINCI.

Comment 4

Some details in the article should be described concisely, and duplicate diagrams, such as those already presented in the main text, do not need to be reiterated in the supplementary materials.

Response to comment 4

We appreciate this important advice. According to your suggestion, we have scrutinized our main text sentence by sentence and went over supplementary materials. We have made our efforts to describe concisely on the details. In this study, we mainly introduced Eu^{3+} and Tb^{3+} for the modulation of RTP performance. The diagrams of Eu^{3+} -doped films are put in the main text while diagrams of Tb^{3+} -doped films are placed in the supplementary materials. The similar spectral features and trends may confuse you and make you feel like they're repetitive.

For example:

In the previous manuscript:

“The 4TPYBOH@PVA films exhibit blue fluorescence with an emission peak at 360 nm (Figure 2b). With the increase of doping concentration, the prompt emission shows a tendency that increased first and then decreased with an optimal luminescence at 0.6

wt%. Notably, the phosphorescence (delayed) spectra reveal dual-band emission with intense blue phosphorescence at around 450 nm and 480 nm, respectively (Figure 2c).”

In the revised version:

“The 4TPYBOH@PVA films exhibit blue fluorescence with an emission peak at 360 nm (Fig. 2b), exhibiting an optimal luminescence at 0.6 wt%. Notably, the phosphorescence (delayed) spectra reveal dual-band emission at around 450 nm and 480 nm (Fig. 2c).”

In the previous manuscript:

“As expected, the Ph-TPY@PVA only displays a dim bluish-green phosphorescence with a relatively shorter duration of about 1s (Figure S6). Also, Ph-TPY@PVA revealed a much shorter lifetime of 0.205 s and an obvious decrement in the intensity of fluorescent and phosphorescence emission (Figures S7 and S8).”

In the revised version:

“As expected, the Ph-TPY@PVA exhibits a degraded optical performance in the fluorescence and phosphorescence emission (Supplementary Figs. 6-8).”

In the previous manuscript:

“Along with increasing the proportion of Tb^{3+} , the fluorescent color gradually changes from red to yellow to opal green, while the phosphorescent performance maintains a relatively stable owing to the constant concentration of total Ln^{3+} (Figure 3b).”

In the revised version:

In the revised version:

“Along with increasing the proportion of Tb^{3+} , the fluorescent color gradually changes from red to yellow to opal green, while the phosphorescent performance maintains unchanged owing to the constant concentration of total Ln^{3+} (Fig. 4b).”

To Reviewer 2

General comments: The manuscript presents a feasible strategy to manipulate RTP performance by dynamic ligand-metal coordination finely. The intensity and duration of afterglow emissions can be well regulated on demand by introducing dynamic coordination to enable efficient energy transfer. Benefiting such modulable performance of these manipulated RTP films, multi-level information encryption including attacker misleading, spatial-time-resolved, and multi-layered cryptographic applications were fully demonstrated. These results sound interesting and can inspire other scientists working in this field. The manuscript is recommended for publication after the following revisions.

Response to the General Comment:

We appreciate your acceptance and recommendation of our work. We also appreciate your positive comments and instructive suggestions on our manuscript. All of your questions have been seriously considered. According to your comments, we carefully made substantial revisions. Many new experimental results and discussions have been added, which have significantly improved our manuscript. These revisions are highlighted with a cyan background in this revised manuscript to facilitate your review. We hope our revised manuscript can be accepted for publication in *Nature Communications*.

Comment 1

The ligand-to-metal photosensitized energy transfer is enabled by the coordination between TPYBOH and lanthanide ions. More detailed experimental data are required to prove the complexity.

Response to comment 1

Thanks a lot for your valuable comment. According to your suggestion, we have performed X-ray photoelectron spectroscopy (XPS) and attenuated total reflectance-Fourier transform infrared (ATR-FTIR) spectroscopy to verify the coordination between 4TPYBOH and lanthanide ions. Both the upshift of the high-resolution N 1s peak and the blue-shift of C=N stretching absorption proved the coordination between TPYBOH and lanthanide ions. The experimental results and discussions are added to the revised manuscript. Please find them in **Paragraph 2 of Page 6 and Supplementary Figs. 10-11** of the revised version.

In the revised manuscript:

As shown in **Fig. 3a** and Supplementary Fig. 9, the red shift and the rise of absorption spectra at 305-380 nm due to the ligand-to-metal charge transfer (LMCT) proves that

more Ln^{3+} were introduced into the organic RTP system with the increase of the immersion concentration.⁵⁶ Moreover, X-ray photoelectron spectroscopy (XPS) and ATR-FTIR spectroscopy were carried out to verify the coordination between TPY subunits and Ln^{3+} . Obviously, the binding energy of N 1s was upshifted from 398.8 eV to 399.6 eV upon introducing Ln^{3+} (Supplementary Fig. 10), attributed to the decrease of electron density on TPY after coordination.⁵⁷ In addition, an evident blue-shift of C=N stretching absorption further proved the coordination between TPYBOH and Ln^{3+} (Supplementary Fig. 11).

In the revised Supporting information:

Supplementary Figure 10. XPS spectra of (a) 4TPYBOH@PVA and (b) 4TPYBOH@PVA-Eu 10. (c) High-resolution XPS spectra (N 1s) of 4TPYBOH@PVA and 4TPYBOH@PVA-Eu 10.

Supplementary Figure 11. ATR-FTIR spectra of PVA, 4TPYBOH@PVA, 4TPYBOH@PVA-Eu 1, and 4TPYBOH@PVA-Eu 10 RTP films.

In the Experimental Section:

Measurements. The spectra of ^1H nuclear magnetic resonance (NMR) was obtained by a Bruker AVANCE 400 MHz spectrometer. X-ray photoelectron spectroscopy (XPS) was carried out with AXIS SUPRA.

Comment 2

The authors should provide the lifetime variations of the fluorescent emission at 360 nm upon increasing the coordination ratio to confirm whether the ligand-to-metal photosensitized energy transfers from the singlet or triplet of the Tpy ligand.

Response to comment 2

Thanks a lot for this valuable comment. According to your advice, we have carefully studied the lifetime variations of the fluorescent emission at 360 nm upon increasing the Ln^{3+} dopant. It was found that fluorescence lifetimes are shortened slightly from 3.07 ns to 2.08 ns upon increasing the Eu^{3+} dopant. To further confirm whether the PSET from the singlet or triplet of the Tpy ligand, triplet quencher cyclooctatetraene (COT) was introduced into the coordination system. Significantly, the emission of $(4\text{TPYBOH})_2\text{-Eu}^{3+}$ was greatly quenched after the addition of triplet quencher cyclooctatetraene (COT), implying that the predominant PSET process should be originated from the triplet of the Tpy ligand. The experimental results and discussions are added to the revised manuscript. Please find them in **Paragraph 2 of Page 8, Paragraph 2 of Page 9, Supplementary Fig. 21 and Supplementary Fig. 27** of the revised version.

In the revised manuscript:

In the Paragraph 2 of Page 8:

Also, the lifetimes are shortened from 0.537 s to 0.307 s for 4TPYBOH@PVA-Eu films and reduced from 0.509 s to 0.185 s for 4TPYBOH@PVA-Tb films, respectively, upon increasing Ln^{3+} concentration from 0.1 to 20 mg/mL. In addition, it is found that the fluorescence lifetimes monitored at 360 nm are shortened slightly from 3.07 ns to 2.08 ns upon increasing the Eu^{3+} dopant (Supplementary Figure 21).

In the Paragraph 2 of Page 9:

Besides, cyclooctatetraene (COT), a well-known triplet quencher,⁶¹ was added to confirm whether the PSET from the singlet or triplet of the organic phosphors. As shown in Supplementary Fig. 27, the emission of $(4\text{TPYBOH})_2\text{-Eu}^{3+}$ was greatly quenched after the addition of triplet quencher COT, implying that the predominant PSET process should be originated from the triplet of the organic phosphors. Furthermore, density functional theory (DFT) calculations were performed to estimate the energy level of the organic phosphor and lanthanide complexes (Supplementary Fig. 28). Compared with organic phosphor, the energy gap of lanthanide complexes was reduced greatly, indicating more easier for the lanthanide complexes to be excited.

In the revised Supporting information:

Supplementary Figure 21. (a) Fluorescence-decay profiles, and (b) Lifetime histogram (monitored at 360 nm) of 4TPYBOH@PVA-Eu films that were prepared by being immersed in different Eu^{3+} aqueous (0.1 to 20.0 mg/mL).

Supplementary Figure 27. Fluorescence spectra of $(4\text{TPYBOH})_2\text{-Eu}^{3+}$ without and with the addition of triplet quencher COT, $\lambda_{\text{ex}} = 254$ nm, numbers represent the volume (unit: μL) of COT additions. The Insets: the photograph of $(4\text{TPYBOH})_2\text{-Eu}^{3+}$ without and with adding 5 μL of COT under 254 nm UV light.

In the Experimental Section:

Materials. Polyvinyl alcohol 1799 (PVA99), $\text{Sm}(\text{NO}_3)_3 \cdot 6\text{H}_2\text{O}$ (99.99%), $\text{Dy}(\text{NO}_3)_3 \cdot 5\text{H}_2\text{O}$ (99.99%), $\text{Pr}(\text{NO}_3)_3 \cdot 6\text{H}_2\text{O}$ (99.99%), $\text{Tm}(\text{NO}_3)_3 \cdot 6\text{H}_2\text{O}$ (99.99%), $\text{La}(\text{NO}_3)_3 \cdot 6\text{H}_2\text{O}$ (99.99%) were purchased from Energy Chemical. $\text{Ho}(\text{NO}_3)_3 \cdot 5\text{H}_2\text{O}$ (99.9%) and cyclooctatetraene (COT) were purchased from Shanghai Macklin Biochemical Co. Ltd.

The solution of $(4\text{TPYBOH})_2\text{-Eu}^{3+}\text{-COT}$: 50.0 mg of 4TPYBOH (0.14 mmol) and 31.6 mg of $\text{Eu}(\text{NO}_3)_3 \cdot 6\text{H}_2\text{O}$ (0.07 mmol) were dissolved in 20.0 mL of DMF to obtain the

homogeneous (4TPYBOH)₂-Eu³⁺ solution. After stirring at room temperature for 1 minute, 2.0 mL of (4TPYBOH)₂-Eu³⁺ solution was taken and mixed with different additions of COT liquid (0.5, 1.0, 2.0, 3.0, 4.0, 5.0 μL) to obtain (4TPYBOH)₂-Eu³⁺-COT. The fluorescence spectra were measured for (4TPYBOH)₂-Eu³⁺ solution and (4TPYBOH)₂-Eu³⁺-COT under 254 nm excitation.

Comment 3

DFT calculations should be provided to estimate the energy level of the organic phosphors and complexes.

Response to comment 3

Thanks a lot for your helpful and suggestive comment. Based on your advice, we have calculated the optoelectronic properties of the organic phosphors and lanthanide complexes in singlet and triplet excited states using density functional theory (DFT) and time-dependent density functional theory (TD-DFT). Compared with organic phosphor, the energy gap of lanthanide complexes was reduced greatly, indicating more easier for the lanthanide complexes to be excited. However, there is no significant difference in the singlet-triplet splitting energy (ΔE_{ST}) for organic phosphors and lanthanide complexes. Please find them in **Paragraph 2 of Page 9 and Supplementary Fig. 28** of the revised version.

In the revised manuscript:

As shown in Supplementary Fig. 27, the emission of (4TPYBOH)₂-Eu³⁺ was greatly quenched after the addition of triplet quencher COT, implying that the predominant PSET process should be originated from the triplet of the organic phosphors. Furthermore, density functional theory (DFT) calculations were performed to estimate the energy level of the organic phosphor and lanthanide complexes (Supplementary Fig. 28). Compared with organic phosphor, the energy gap of lanthanide complexes was reduced greatly, indicating more easier for the lanthanide complexes to be excited. However, there is no significant difference in the singlet-triplet splitting energy (ΔE_{ST}) for organic phosphors and lanthanide complexes. Through the above analysis, the PSET from the triplet state of 4TPYBOH phosphor to Ln³⁺ should be a dominant pathway.

In the revised Supporting information:

Supplementary Figure 28. The DFT calculation of the organic phosphors and lanthanide complexes. (a,b) Calculated frontier molecular orbitals and orbital energies of 4TPYBOH and $(4TPYBOH)_2-Eu^{3+}$, respectively. (c,d) Vertical excitation energy levels of 4TPYBOH and $(4TPYBOH)_2-Eu^{3+}$, respectively.

In the Experimental Section:

Theoretical Calculations. The ground and excited-state optimizations were calculated using the Density Functional Theory (DFT) and time-dependent Density Functional Theory (TD-DFT) at the PBE0-D3¹/6-311G(d,p)² + SDD³ level with Gaussian 16, respectively. Here, the SDD effective core potential was used to describe the atomic orbital and relativistic effect of the heavy element Europium.

Comment 4

Eu^{3+} and Tb^{3+} metal ions were utilized to manipulate the RTP properties. Can the RTP properties also be manipulated by the other lanthanide ions? What happens if other lanthanide ions such as La^{3+} and Tm^{3+} are applied for complexation?

Response to comment 4

Thank you for your suggestive comments. According to your suggestion, we have also studied the optical modulation by introducing other lanthanide metal ions including La^{3+} , Sm^{3+} , Dy^{3+} , Pr^{3+} , Ho^{3+} and Tm^{3+} . It's found that Ln^{3+} ions with efficient PSET are highly desirable for finely manipulating RTP properties, except for different abilities in the modulation of prompt emission. The corresponding fluorescence spectra, phosphorescence spectra, photographs and necessary discussions have been fully added

to the revised manuscript. Please find them in **Paragraph 3 of Page 8, Paragraph 2 of Page 10, Supplementary Fig. 23** and **Supplementary Fig. 33** of the revised version.

In the revised manuscript:

In Paragraph 3 of Page 8:

Moreover, La^{3+} -doped RTP films were prepared and studied considering that the La^{3+} had no available energy levels for energy transfer.^{59, 60} As expected, the RTP intensity and duration of delayed emission remain nearly constant (Supplementary Fig. 23), although an increase of La^{3+} dopants may slightly weaken the hydrogen bonding (Supplementary Fig. 24).

In Paragraph 2 of Page 10:

To investigate whether the optical properties could be manipulated by the other lanthanide metals, we further introduced Sm^{3+} , Dy^{3+} , Pr^{3+} , Ho^{3+} , and Tm^{3+} into the pristine organic RTP system. As shown in Supplementary Fig. 33, the RTP emission can be also finely modulated by doping these Ln^{3+} , in addition to relatively smaller changes in fluorescence emission compared with Eu^{3+} - and Tb^{3+} -doped films. On the one hand, phosphorescence spectra of these Ln^{3+} -manipulated RTP films all exhibit characteristic emission peaks at 450 nm and 480 nm, accompanied by significant decreases in the intensity and duration of RTP emission upon increasing the amounts of Ln^{3+} dopants. It indicates that there is an efficient PSET between organic phosphors and Ln^{3+} . On the other hand, obvious characteristic fluorescence peaks of Sm^{3+} and Dy^{3+} emission can be observed, especially for Sm^{3+} -doped film that exhibit crimson fluorescence. However, no distinct fluorescence peaks of Pr^{3+} , Ho^{3+} , and Tm^{3+} can be detected, which is attributed to the small energy gap between their excited and ground states, and the denser energy levels lead to severe non-radiative transition and energy dissipation.^{62, 63} Taken together, Ln^{3+} ions with efficient PSET are expected to finely manipulate RTP properties, except for different abilities in the modulation of prompt emission.

In the revised supporting information:

Supplementary Figure 23. (a) The photographs of La^{3+} -doped RTP films (prepared by soaking 4TPYBOH@PVA in 1.0 and 10.0 mg/mL of La^{3+} aqueous solutions), the photos were taken

under 254 nm UV excitation and at different time intervals after the removal of UV irradiation.

(b) Phosphorescence spectra of corresponding La^{3+} -doped RTP films.

Supplementary Figure 33. (a) The fluorescence and (b) phosphorescence spectra of Sm^{3+} -, Dy^{3+} -, Pr^{3+} -, Ho^{3+} -, and Tm^{3+} -doped RTP films that were prepared by being immersed in La^{3+} aqueous solution (1.0 and 10.0 mg/mL). (c) The photographs of Sm^{3+} -, Dy^{3+} -, Pr^{3+} -, Ho^{3+} -, and Tm^{3+} -doped RTP films that were prepared by being soaked in La^{3+} aqueous solution (1.0 and 10.0 mg/mL) taken under and after removing the 254 nm UV irradiation.

In the Experimental Section:

Materials. Polyvinyl alcohol 1799 (PVA99), $\text{Sm}(\text{NO}_3)_3 \cdot 6\text{H}_2\text{O}$ (99.99%), $\text{Dy}(\text{NO}_3)_3 \cdot 5\text{H}_2\text{O}$ (99.99%), $\text{Pr}(\text{NO}_3)_3 \cdot 6\text{H}_2\text{O}$ (99.99%), $\text{Tm}(\text{NO}_3)_3 \cdot 6\text{H}_2\text{O}$ (99.99%), $\text{La}(\text{NO}_3)_3 \cdot 6\text{H}_2\text{O}$ (99.99%) were purchased from Energy Chemical. $\text{Ho}(\text{NO}_3)_3 \cdot 5\text{H}_2\text{O}$ (99.9%) and cyclooctatetraene (COT) were purchased from Shanghai Macklin Biochemical Co. Ltd.

4TPYBOH@PVA-Eu/Tb films: First, $\text{Eu}(\text{NO}_3)_3 \cdot 6\text{H}_2\text{O}$ or $\text{Tb}(\text{NO}_3)_3 \cdot 5\text{H}_2\text{O}$ was added to H_2O to prepare the aqueous solutions with a concentration gradient (0.1 mg/mL, 0.5 mg/mL, 1.0 mg/mL, 3.0 mg/mL, 5.0 mg/mL, 10.0 mg/mL, 20.0 mg/mL). Then, the pristine RTP (0.6 wt% 4TPYBOH@PVA) films were immersed in different Eu^{3+} or Tb^{3+} aqueous for 5 minutes and heated at 65 °C for 2 hours to remove H_2O . Finally, we can obtain the Ln^{3+} -doped RTP film with different doping amounts. Sm^{3+} -, Dy^{3+} -, Pr^{3+} -, Ho^{3+} -, Tm^{3+} - and La^{3+} -doped RTP films were prepared by the same method with 4TPYBOH@PVA-Eu/Tb as described above.

Comment 5

In the demonstration part, the authors present several creative applications in multi-level information encryption and anti-counterfeiting. More discussions are needed on the advantages of such fluorescence-phosphorescence dual-mode encryption over the other optical encryption strategies.

Response to comment 5

Thanks a lot for your helpful and inspiring comments. Based on your suggestion, we have added more discussions on the advantages of such fluorescence-phosphorescence dual-mode encryption over the other optical encryption strategies. Please find them in **Paragraph 2 of Page 13** of the revised manuscript:

In the revised manuscript:

Based on the above-established multi-level information encryption, we further developed a virtually unbreakable multi-layer encryption system for important information storage (**Fig. 6a**). Different from the reported encryption strategies, we developed an innovative method for camouflaging information by multicolour fluorescence and decryption in the form of delayed emission, yielding greatly improved security. Moreover, the important data can be further encrypted in a spatial-time-resolved way based on the finely modulated RTP system. In this way, the information storage capacity is greatly enlarged and the important data can be hidden in a multi-level way with a specific code for each layer, greatly reducing the risk of being hacked. As a proof-of-concept, we designed and fabricated an encrypted dot matrix (8×5) by using a series of 4TPYBOH@PVA-Eu and 4TPYBOH@PVA-Tb RTP materials (Supplementary Fig. 37).

To Reviewer 3

General comments: The manuscript (Ref: NCOMMS-23-56012) of the paper entitled "Finely Manipulating Room Temperature Phosphorescence by Dynamic Lanthanide Coordination toward Multi-Level Information Security ", in my opinion can be considered for publication in the journal: "Nature Communications" after its minor revision. The article shows very interesting and novel results concerning new the use of organic lanthanide-based phosphorescent materials for real-world applications including anti-counterfeiting and optical coding. The work is very well executed, the results are novel and reliable (the postulated effects are well-proved), and the presented findings deserve to be published in this journal. However, before publication of the manuscript, the following points should be addressed:

Response to the General Comment:

We are very grateful to you for your acceptance and recommendation of our work. We also appreciate your positive comments and many instructive suggestions. All of your questions have been seriously considered. According to your suggestive comments, we have corrected all mistakes and added detailed discussions, which have significantly improved our manuscript. These revisions are highlighted with a green background in this revised manuscript to facilitate your review. We hope our revised manuscript can be accepted by *Nature Communications*.

Comment 1

- the Authors use the phrase "afterglow" thorough the manuscript, however, this expression is reserved for inorganic phosphors exhibiting emission from the traps (stimulated by temperature), and it is frequently mistakenly used for organic phosphorescent materials, which exhibit just long lasting emission due to the spin-forbidden character of the radiation processes – triplet-singlet emission (see the fundamental work of professor Hölsä, J. Persistent Luminescence Beats the Afterglow: 400 Years of Persistent Luminescence. Electrochem. Soc. Interface 2009, 18 (4), 42–45. <https://doi.org/10.1149/2.F06094IF>.)

Response to comment 1

Thank you for this important question and patient explanation. We have carefully read the reference that you provided. According to your suggestion, we have corrected the phrase "afterglow" to "long-lived emission", "phosphorescence" or "phosphorescent" in our revised version.

For example:

In the revised manuscript:

“Room temperature phosphorescence (RTP) is a fascinating optical phenomenon in which **long-lived emission** can last from several seconds to hours under ambient conditions after removing the excitation light source.^{1-5”}

“Compared with the most widely used fluorescent tags for information encryption,²⁴⁻²⁸ RTP materials exhibit highly concealable and unclonable for innovative multi-level information coding owing to their additional time dimension and tunable optical properties (**phosphorescence** lifetime, color, and intensity).^{29-35”}

“Despite the impressive advancements that have been made, the on-demand manipulation of RTP properties to achieve desirable **phosphorescent** performance without any tedious preparation process remains a formidable challenge, which greatly limits its related applications in high-level information encryption and anti-counterfeiting.”

Comment 2

- the Authors write about the application of room-temperature phosphorescence for information encryption, but it is suggested to write something/compare with encryption and anti-counterfeiting with afterglow emission of lanthanide and d-block metal ions (see e.g. 10.1002/adfm.202307791; 10.1002/adom.202300600)

Response to comment 2

Thanks a lot for your helpful comments. We have carefully read the related paper that you recommended about encryption and anti-counterfeiting with afterglow emission of lanthanide and d-block metal ions. According to your suggestions, we have added more discussions on the comparison between our encryption system and those encryption with afterglow emission of lanthanide and d-block metal ions. Besides, we have added the related references in the revised version. Please find them in **Paragraph 1 of Page 2, Paragraph 2 of Page 13 and references 36-39** of the revised manuscript:

In the revised manuscript:

In the Paragraph 1 of Page 2:

Compared with the most widely used fluorescent tags for information encryption,²⁴⁻²⁸ RTP materials exhibit highly concealable and unclonable for innovative multi-level information coding owing to their additional time dimension and tunable optical properties (**phosphorescence** lifetime, color, and intensity).²⁹⁻³⁵ **In recent years, inorganic afterglow materials with multimodal excitation properties have demonstrated the advantages of encryption and anti-counterfeiting with multi-dimensional**

information.³⁶⁻³⁹ However, inorganic afterglow materials are mainly prepared by ion doping, suffering from a high preparation temperature, environmental pollution, and limited metal resources. To address these issues, researchers have turned to organic RTP materials for developing advanced information encryption.

In the Paragraph 2 of Page 13:

Based on the above-established multi-level information encryption, we further developed a virtually unbreakable multi-layer encryption system for important information storage (**Figure 6a**). Different from the reported encryption strategies, we developed an innovative method for camouflaging information by multicolour fluorescence and decryption in the form of delayed emission, yielding greatly improved security. Moreover, the important data can be further encrypted in a spatial-time-resolved way based on the finely modulated RTP system. In this way, the information storage capacity is greatly enlarged and the important data can be hidden in a multi-level way with a specific code for each layer, greatly reducing the risk of being hacked. As a proof-of-concept, we designed and fabricated an encrypted dot matrix (8×5) by using a series of 4TPYBOH@PVA-Eu and 4TPYBOH@PVA-Tb RTP materials (Supplementary Fig. 37).

References

35. Ma, L. et al. A Universal Strategy for Tunable Persistent Luminescent Materials via Radiative Energy Transfer. *Angew. Chem. Int. Ed.* **61**, e202115748 (2022).
36. Runowski, M. et al. Multimodal Optically Nonlinear Nanoparticles Exhibiting Simultaneous Higher Harmonics Generation and Upconversion Luminescence for Anticounterfeiting and 8-bit Optical Coding. *Adv. Funct. Mater.* **34**, 2307791 (2024).
37. Xue, J. et al. Precisely Manipulating the Self-Reduction of Manganese in MgGa₂O₄ through Lithium Incorporation for Optical Thermometry and Anti-Counterfeiting. *Adv. Opt. Mater.* **11**, 2300600 (2023).
38. Pei, P., Bai, Y., Su, J., Yang, Y. & Liu, W. Achieving mechano-upconversion-downshifting-afterglow multimodal luminescence in a lanthanide-doped LaCaAl₃O₇ phosphor for multidimensional anticounterfeiting. *Sci. China Mater.* **65**, 2809-2817 (2022).
39. Lei, L. et al. Manipulation of time-dependent multicolour evolution of X-ray excited afterglow in lanthanide-doped fluoride nanoparticles. *Nat. Commun.* **13**, 5739 (2022).
40. Wei, P. et al. New Wine in Old Bottles: Prolonging Room-Temperature Phosphorescence of Crown Ethers by Supramolecular Interactions. *Angew. Chem. Int. Ed.* **59**, 9293-9298 (2020).

Comment 3

- in the next paragraph, the Authors write about the use of lanthanide ions in information encryption and optical sensors, but they disregarded the recent breakthrough achievements in this field (e.g. 10.1002/adma.202302749; 10.1002/adfm.202214663; 10.1002/adfm.202307791

Response to comment 3

We apologize for missing the recent breakthrough achievements in this field. We have cited the corresponding literature as **references 36, 47 and 48** in the introduction.

In the revised manuscript:

Lanthanide (Ln^{3+}) metal ions such as Eu^{3+} and Tb^{3+} as iconic luminescence centers have been widely used for practical applications in the field of emissive sensors, information encryption, and anti-counterfeiting.^{36, 42-48} Since the f-f transition of Ln^{3+} species is a forbidden process, it is generally necessary to select organic ligands with high molar coefficients for sensitization to realize highly efficient emission, which is also called “antenna effect”.^{49, 50}

References

36. Runowski, M. et al. Multimodal Optically Nonlinear Nanoparticles Exhibiting Simultaneous Higher Harmonics Generation and Upconversion Luminescence for Anticounterfeiting and 8 - bit Optical Coding. *Adv. Funct. Mater.* **34**, 2307791 (2024).
47. Zheng, T. et al. Giant Pressure - Induced Spectral Shift in Cyan - Emitting Eu^{2+} - Activated $\text{Sr}_8\text{Si}_4\text{O}_{12}\text{Cl}_8$ Microspheres for Ultrasensitive Visual Manometry. *Adv. Funct. Mater.* **33**, 2214663 (2023).
48. Brites, C. D. S. et al. Spotlight on Luminescence Thermometry: Basics, Challenges, and Cutting-Edge Applications. *Adv. Mater.* **35**, e2302749 (2023).

Comment 4

- the Author write: “excited singlet state (S_1) of Ln^{3+} ” – which is incorrect in relation to most if Ln^{3+} ions. This is because, as in the case of the radiative transitions within Ln^{3+} ions, occur usually between the terms of higher multiplicity (including tetraplets, pentaplets, hexaplets, and so on) with different total spin momentum (forbidden 4f-4f transitions), and such radiative processes should be called luminescence. so more correctly, would be to say: “higher excited states of Ln^{3+} ” or “excited states of Ln^{3+} with higher multiplicity”.

- The above mentioned mistake also includes the energy level diagram in Scheme 1.

Response to comment 4

Thank you very much for carefully pointing out this problem. According to your suggestion, we have revised the “excited singlet state (S_1) of Ln^{3+} ” to “higher excited states of Ln^{3+} ”, please find them in **Paragraph 2 of Page 2** of the revised manuscript. Also, we have corrected this mistake in the energy level diagram of Fig. 1. Please find them in **Fig. 1a** of the revised version.

In the revised manuscript:

The ligand-to-metal photosensitized energy transfer (PSET) process originates from the triplet state (T_1) of organic ligand to **the higher excited states of Ln^{3+}** ,⁵¹⁻⁵³ which is a competing pathway for the radiation transition of organic triplet-state exciton to the ground state.

Fig. 1 The illustration of manipulating RTP properties and multi-level information encryption. **a** The introduction of Ln^{3+} (Eu^{3+} and Tb^{3+}) into PVA-terpyridine phenylboronic acids (TPYBOH@PVA) for finely manipulating RTP properties via the ligand-to-metal photosensitized energy transfer process, the resulting Ln^{3+} -manipulated materials showing multicolor fluorescence and mutual blue phosphorescence. **Ex. = excitation, Fluor. = fluorescence, Phos. = phosphorescence, Lumin. = luminescence.** **b** The proof-of-concept demonstrations of the advanced multi-level encryption and anti-counterfeiting based on Ln^{3+} -manipulated RTP materials, revealing greatly enhanced security level.

Comment 5

Page 4; line 97-98; the Authors write: “which is rarely reported but appealing for high-level information encryption.” – provide the missing reference

Response to comment 5

We apologize for missing the citation of related reviews. We have cited the corresponding literature as **references 41 and 54** in the introduction.

In the revised manuscript:

Moreover, the Ln^{3+} -manipulated RTP system (TPYBOH@PVA-Ln) reveals illusive long-wavelength fluorescence emission and short-wavelength phosphorescence emission, which is rarely reported but appealing for high-level information encryption.^{41, 54}

References

41. Lin, F. et al. Stepwise Energy Transfer: Near-Infrared Persistent Luminescence from Doped Polymeric Systems. *Adv. Mater.* **34**, 2108333 (2022).
54. Yu, X. et al. Ln-Mof-Based Hydrogel Films with Tunable Luminescence and Afterglow Behavior for Visual Detection of Ofloxacin and Anti-Counterfeiting Applications. *Adv. Mater.*, e2311939 (2024).

Comment 6

- “As shown in Figure 2a and S9, the red shift and the rise of absorption spectra at 305-380 nm due to the charge transfer proves that more Ln^{3+} were introduced into the organic RTP system with the increase of the immersion concentration.” – I think that the authors meant “energy transfer” instead of “charge transfer”

Response to comment 6

Thank you for this comment. We should explain the red shift of absorption spectra more clearly. Thus, we have added more detailed explanation on the variation of absorbance spectra in our revised version. The ligand-to-metal charge transfer (LMCT) often brings the bathochromic shift of absorbance and the emergence of tailed regions (J. Chem. Phys. 2005, 122, 54109). Thus, the red shift and the rise of absorption spectra at 305-380 nm can be attributed to LMCT upon increasing Ln^{3+} dopants. Please find them in **Paragraph 2 of Page 6** of the revised manuscript:

In the revised manuscript:

As shown in **Fig. 3a** and Supplementary Fig. 9, the red shift and the rise of absorption spectra at 305-380 nm due to the **ligand-to-metal charge transfer (LMCT)** proves that more Ln^{3+} were introduced into the organic RTP system with the increase of the immersion concentration.⁵⁶

Comment 7

- “long-lived Ln³⁺ luminescence with lifetime at tens of milliseconds” - lifetimes of some Ln³⁺ (including Eu³⁺ and Tb³⁺) in inorganic matrices are typically within few ms, and within hundreds of μs for organic compounds. From the decay curves in Figure S15, it is clear that the mentioned lifetimes are about 1 ms (not tens of ms)

Response to comment 7

Thanks a lot for pointing out this important problem and providing a patient explanation. We are sorry for the misleading descriptions of the lifetime of Ln³⁺. According to your suggestions, we have revised the mistakes. Please find them in **Paragraph 2 of Page 8** of the revised manuscript.

In the revised manuscript:

In phosphorescence spectra, the characteristic peaks of Eu³⁺ and Tb³⁺ still can be observed with a delay time of 0.05 ms (Supplementary Fig. 16), due to the long-lived Ln³⁺ luminescence with lifetimes at **hundreds of microseconds** (Supplementary Fig. 17), which is hardly observable to the naked eye.

Comment 8

- the Authors write: “luminescent lifetimes” but the correct expression is “excited state lifetime” or “luminescence decay time”

- change: “phosphorescence emission is always larger than the fluorescence emission.” To “phosphorescence emission is always longer than the fluorescence emission.”

- change: “phosphorescent spectra (emission)” to “phosphorescence spectra (emission)”

- change: “lifetimes are decreased” to “lifetimes are shortened”

Response to comment 8

Thank you very much for carefully pointing out these issues. We are sorry for providing the inappropriate expressions in the manuscript. According to your suggestions, we have carefully examined the full text and revised all the mistakes.

In the revised manuscript:

In Paragraph 1 of Page 2:

“Room temperature phosphorescence (RTP) is a fascinating optical phenomenon in which long-lived emission can last from several seconds to hours under ambient conditions after removing the excitation light source.¹⁻⁵ Due to its unique luminescence properties such as long **luminescence decay lifetime** and large Stokes shift,⁶⁻¹¹ it has

become a promising candidate for numerous applications, such as information encryption,¹²⁻¹⁶ bioimaging,¹⁷⁻²⁰ and organic optoelectronics.²¹⁻²³”

In Paragraph 1 of Page 3:

“In addition, according to the RTP emission mechanism, the wavelength of phosphorescence emission is always longer than the fluorescence emission.”

In Paragraph 2 of Page 8:

“In phosphorescence spectra, the characteristic peaks of Eu^{3+} and Tb^{3+} still can be observed with a delay time of 0.05 ms (Supplementary Fig. 16), due to the long-lived Ln^{3+} luminescence with lifetime at hundreds of microseconds (Supplementary Fig. 17), which is hardly observable to the naked eye.”

In Paragraph 2 of Page 8:

“That is, the phosphorescent properties are highly dependent on the amount of introduced Ln^{3+} . Also, the lifetimes are shortened from 0.537 s to 0.307 s for 4TPYBOH@PVA-Eu films and reduced from 0.509 s to 0.185 s for 4TPYBOH@PVA-Tb films, respectively, upon increasing Ln^{3+} concentration from 0.1 to 20 mg/mL.”

REVIEWERS' COMMENTS

Reviewer #1 (Remarks to the Author):

The revised manuscript carefully addresses my general comments, and the quality is greatly improved. Regarding my concerns about the novelty, I hold my opinion. However, the entire investigation is highly valuable to the field and can be accepted.

Reviewer #2 (Remarks to the Author):

The previous reviewers have provided numerous constructive comments, and the authors have made revisions and additions in response to these suggestions. The current version appears to be more publishable. Therefore, I recommend an acceptance of the manuscript.

Reviewer #3 (Remarks to the Author):

The Authors corrected the manuscript according to the Reviewers` comments, so the manuscript can be published in the present form.

Point-by-point response to the comments

To Reviewer 1

Comments: The revised manuscript carefully addresses my general comments, and the quality is greatly improved. Regarding my concerns about the novelty, I hold my opinion. However, the entire investigation is highly valuable to the field and can be accepted.

Response: We are grateful for your kind recommendation!

To Reviewer 2

Comments: The previous reviewers have provided numerous constructive comments, and the authors have made revisions and additions in response to these suggestions. The current version appears to be more publishable. Therefore, I recommend an acceptance of the manuscript.

Response: We appreciate your acceptance and recommendation of our work!

To Reviewer 3

Comments: The Authors corrected the manuscript according to the Reviewers' comments, so the manuscript can be published in the present form.

Response: We thank you for supporting the publication of this work in Nature Communications!